# Detection of Rare Objects by Flow Cytometry: Imaging, Cell Sorting, and Deep Learning Approaches

**DOI:** 10.3390/ijms21072323

**Published:** 2020-03-27

**Authors:** Denis V. Voronin, Anastasiia A. Kozlova, Roman A. Verkhovskii, Alexey V. Ermakov, Mikhail A. Makarkin, Olga A. Inozemtseva, Daniil N. Bratashov

**Affiliations:** 1Laboratory of Biomedical Photoacoustics, Saratov State University, 410012 Saratov, Russia; 2Department of Physical and Colloid Chemistry, National University of Oil and Gas (Gubkin University), 119991 Moscow, Russia; 3School of Urbanistics, Civil Engineering and Architecture, Yuri Gagarin State Technical University of Saratov, 410054 Saratov, Russia; 4Department of Biomedical Engineering, I. M. Sechenov First Moscow State Medical University, 119991 Moscow, Russia

**Keywords:** imaging flow cytometry, cell sorting, deep learning, circulating tumor cells, cell labeling, liquid biopsy, flow cytometry data analysis

## Abstract

Flow cytometry nowadays is among the main working instruments in modern biology paving the way for clinics to provide early, quick, and reliable diagnostics of many blood-related diseases. The major problem for clinical applications is the detection of rare pathogenic objects in patient blood. These objects can be circulating tumor cells, very rare during the early stages of cancer development, various microorganisms and parasites in the blood during acute blood infections. All of these rare diagnostic objects can be detected and identified very rapidly to save a patient’s life. This review outlines the main techniques of visualization of rare objects in the blood flow, methods for extraction of such objects from the blood flow for further investigations and new approaches to identify the objects automatically with the modern deep learning methods.

## 1. Introduction

The problem of detection and extraction of rare objects from the blood flow arises in a number of situations. This includes the search for the very rare circulating tumor cells (CTCs) at early stages of cancer development by liquid biopsy [1,2], the detection of microorganisms during acute blood infections to determine its strain very rapidly [3], early detection of malaria parasites including in vivo [4,5] and other pathogenic states that impose high risks to human life and well-being. In addition to the detection and extraction of such rare objects, a lot of developments are targeted on eliminating it from the blood flow by a sort of blood filtering. The current state of the art in this field is defined by the progress in cell imaging and sorting techniques, sample enrichment, and separation along with the new approaches for automatization of data analysis based on machine learning and deep learning methods. Here we provide an overview of different techniques designed to detect very rare objects in the blood flow, sort it out or filter from the bloodstream and extract for further investigation.

According to the statistics of the World Health Organization for 2016, around 71% of the overall 57 million deaths are caused by noncommunicable diseases, including cardiovascular disease—31% and cancer—16% [6]. During 2016, 216 million cases of malaria were detected [7]. The promising approach to diagnose these diseases is the detection of untypical objects in blood and lymph samples. However, there are two significant challenges: the first one is the rarity of untypical blood elements [8] and the second one is the small volume of the sample [9] that has to be subjected to detailed analysis. The last generation of the flow cytometry systems with the possibility of object visualization allows verifying received data but has additional restrictions. For example, the object size is limited by the resolution of the cytometer optical system, and there are flow speed limits based on the sensitivity of detection scheme, laminarity of object flow and required quality of the image. However, there is no doubt that a solution to the current main medical issues is related to the detailed analysis of single cells, which requires their separation from ordinary objects.

Currently, liquid biopsy is one of the most informative and broadly described analyses in medicine. A number of diseases could be diagnosed by detecting of untypical blood objects—emboli in the lymph or blood. For example, CTCs are the prognostic factor of different cancer types [10,11,12,13,14,15,16], which can be detected at early stages. Some species of Protozoa that parasitize in human (e.g., *Babesia microti*, *B. divergens* [17], *Trypanosoma cruzi* [18], *Plasmodium falciparum*, *P. vivax*, *P. ovale*, *P. malariae*, *P. knowlesi* [19], *T. brucei* [20]) and cause infectious diseases like babesiosis [17], Chagas disease [18], malaria [19], African trypanosomiasis [20], etc. can be detected in whole blood. Bloodstream infections [21] also can be diagnosed by the detection of microorganisms in the blood [22]. Theoretically, other emboli types such as a fat embolism [23] and blood clots [24] can be found by the analysis of a blood sample as well.

Generally, there are two opposite approaches for the selection of target cell subpopulations from the entire population. The positive selection implicates the direct isolation of target objects from a general population. Oppositely, the negative selection means the exclusion of all objects except for the target [25]. Both of these methods have advantages and disadvantages. However, the negative approach is more efficient for untypical object analysis in lymph or blood due to the exclusion of all objects except for embolus.

The significant step towards the isolation of rare blood circulating objects was the invention of the Fluorescence Activated Cell Sorter (FACS) by Bonner, Sweet, Hulett, Herzenberg et al. in the 60s of the last century [26]. Development of new fluorophores and methods of labeling different cell structures allowed for sorting cells according to many features and selection of small subpopulations and even single cells [27]. Currently, there are a number of methods based on the physical and biological properties of cells, allowing their sorting.

Here we review the modern methods and approaches used for flow cytometer design, cell labeling, their viability evaluation, and cell sorting along with other methods to separate cell subpopulations and the automatic approaches for following data analysis based on machine learning and deep learning methods.

## 2. Flow Cytometry Hardware

The optical detection system is the main part of the flow cytometer that define the overall system performance and provide the quality of data (high signal-to-noise ratio, high sensitivity, good repeatability) at a reasonable processing speed. Typically, a flow cytometry system consists of three main parts: illumination subsystem, usually including one or multiple lasers of different wavelengths; fine-tuned optics, comprising dichroic band-pass and cut-off filters; and detection system, usually based on high-sensitivity photomultiplier tubes (PMTs) or camera for imaging systems.

### 2.1. Illumination Subsystem

Lasers are the excitation light sources for virtually every modern flow cytometer. They should provide stable, monochromatic, coherent light for both forward- and side scatter channels of detection as well as to excite various fluorescent probes containing in cells to identify them and to investigate their morphology, cell cycle state, etc. [28] Although the first cytometers were based on lamp sources like mercury lamps, with the technology development they were replaced by the lasers due to their higher stability and the ability to produce highly coherent light.

About 40 years have gone since the creation of the first 488 nm laser, nevertheless, blue-green argon-ion lasers are still the most frequently used because of the high variety of fluorescent labels excited at this wavelength: fluorescein, acridine, and their derivatives, cell viability dyes Calcein AM and propidium iodide, etc. [29] However, with the development of cytometry, the number of new fluorochromes increased, which caused further production of lasers with different wavelengths, from ultraviolet to infrared. Currently, the excitation of almost full UV–visible spectrum is provided by the combination of earlier gas sources and modern solid-state lasers [30]. Nevertheless, the combination of only three of them (ultraviolet, 488 nm, and red diode) in one flow cytometer could provide theoretically the ability to analyze up to 17 existing fluorescent labels and could also give access to fluorochromes previously unavailable on usual instruments. The employment of additional lasers, in turn, can increase the number of simultaneously measured parameters, so advanced flow cytometers support the introduction of up to 10 lasers with different wavelengths to maximize sensitivity and allow tuning of excitation conditions to the precise experiments.

#### 2.1.1. Laser Separation

The choice of the laser for each cytometer is limited by a number of technical parameters that should be taken into account. First, the two or more excitation sources used in one flow cytometer must be separated to allow the interrogation of cells and stream by multiple lasers. In this case spatially and temporally separated laser beams could be used: each laser focuses on its own point of the stream when appropriate collection pinholes are aligned to the particular part of the flow channel. This scheme can include seven or more lasers and collection channels simultaneously.

Another separation way is used in commercial cytometers like Accuri™/Accuri™ Plus (BD Biosciences, San Jose, CA, USA) and Guava^®^easyCyte™ (Luminex, Austin, TX, USA). There are several collinear lasers, temporally separated to avoid overlapping of the fluorochrome emission and the laser excitation light at the same wavelength, for example, in a combination of PE-Cy5 dye and red HeNe laser. Here the subsequent picosecond activation of each laser is used with a synchronized simultaneous activation of the corresponding detector.

#### 2.1.2. Laser Type

There are two main groups of lasers integrated into flow cytometers, which are the earlier generation of gas lasers (water-cooled and then air-cooled) and modern solid-state lasers. The first produced 488 nm laser had the argon-ion active medium and water cooling. Since it could produce only a single wavelength excitation, more advanced cell sorters were equipped by krypton-ion lasers, which were able to generate high power signals up to several hundred mW at a number of various wavelengths. Further, the water-cooled lasers were replaced by air-cooled, which, however, have a lack of high-power output signal (10–30 mW). One of them, helium-neon (HeNe), took its place in many flow cytometers by generating an important red 633 nm laser line along with less powerful green (543 nm), yellow (592 nm), and orange (612 nm) lines.

After the development of modern solid-state light sources: laser diodes and diode-pumped solid-state (DPSS) lasers they have mostly replaced the gas lasers. Direct diodes use a semiconductor heterostructure to directly produce a laser line when DPSS source pumps a solid-state nonlinear medium by an infrared laser and generates output wavelength by doubling, tripling and quadrupling of the initial one.

#### 2.1.3. Beam Shape and Quality

Beam shape, quality, and laser noise level are critical parameters for virtually any flow cytometer. There are two types of laser beam profiles: a single-mode having a single circular or elliptical spot and a multi-mode providing multiple spots along one optical axis. Typically flow cytometers require a single-mode beam with a circular Gaussian transverse electronic mode (TEM_00_) characteristic. Some flow cytometers can be equipped with optics allowing for an additional reshaping of the beam into an elliptical one. A single spot is easily focused on the stream while mixing of prisms and cylindrical lenses can provide a flat-top profile with sharper edges of the beam. This mode is excellently generated by gas lasers and DPSS laser sources.

The multi-mode beam generating by a direct diode is more powerful but has a complex beam profile with several peak patterns. In this form, it can not be used in flow cytometry as complex beam handling may cause large power losses. However, the beam profile can be significantly improved if passed through cylindrical lenses or anamorphic prism pairs.

Laser noise level refers to the degree of its stability in the time interval the cell passes through the exposed region. It should not exceed 0.5% peak-to-peak signal amplitude and can vary from 20 kHz to 20 MHz with respect to many factors, including the laser medium, power supply, electronics, and laser self-regulation [30]. Although gas lasers and diodes have self-regulating light control mode, for the DPSS and some diode sources the temperature control is a critical issue and can influence the resulting laser noise. It also limits the applications of the particular laser because of the decrease in resolution with noise growth. However, modern low-noise flow cytometers allow us to carry out successfully precise measurements like DNA analysis and immunophenotyping.

One more important parameter is laser collimation, corresponding to the degree of divergence of the laser beam after it leaves the output. Currently available commercial sources integrated into flow cytometers provide a well-collimated beam even several meters from the output that is sufficient for their further focusing and shaping.

#### 2.1.4. Laser Delivery

The delivery of laser beam to the stream and light intersect region can occur in two ways: in free space, using prisms, dichroic mirrors, and long-pass filters to merge and steer multiple beams and focus them on the stream; and in single-mode fiber optics. Using fiber does not require any additional expensive optic elements, provides alignment stability and is generally safer in comparison to free space sources. Any defects in the beam profile can be fixed by cutting the edges of spatial distribution, however, this leads to losses in the resulting power. The FACSAria™ (BD Biosciences) cytometer uses a fiber system for laser delivery.

However, there are some disadvantages including significant power losses for UV and violet laser wavelengths and the expensive fiber materials, degrading with time. Therefore, modern flow cytometers like Amnis^®^ ImageStream^®X^ Mark II (Luminex) and CytoFLEX^®^ (Beckman Coulter, Brea, CA, USA) system still include free-space laser delivery. Additionally, high-quality damage-resistant coated dichroic mirrors are required to work in the UV range.

#### 2.1.5. Laser Coherence

The spectral width of the line that laser emits is a critical parameter as well. This value does not exceed 1 nm for highly coherent lasers, although the light, produced by the laser diodes, has the main intensive part at one wavelength and slightly less intensive background at lower and higher wavelengths. For the application of these lasers, modern cytometers are equipped with additional narrow width notch filters that make laser line essentially monochromatic.

### 2.2. Optical Arrangement

Since modern cytometers provide both brightfield and fluorescence imaging of flowing objects, they provide identification of the single cell according to the distribution of fluorescent markers and the cell morphology. However, there are several optical schemes for visualization of flowing objects in real-time.

The first kind of imaging optical system is realized in the Amnis^®^ (Luminex) imaging flow cytometer, which was designed to enhance the fluorescent sensitivity in comparison to fluorescent microscopes and provide high-speed operation comparable with conventional flow cytometers (up to several thousands of objects per second). The basic scheme of the optical system is presented in Figure 1. The light sheet illumination of cells flowing one-by-one through the illumination region provides scattered light (forward and side channel) and the set of fluorescent channels with different excitation source and detection beam path. Diffusion of light at small angles (forward scattered channel, FSC), where the light is collected between 0.5–8° to the optical axis, is used to determine the cell size [31]. At the side scatter channel (SSC), the light is collected orthogonally to the optical axis and represents a combination of the diffusion, reflection, and refraction components that are related to cell morphology, granulation, and other structural properties. A high numerical aperture objective lens collects either emitted fluorescence, scattered and transmitted light from the cells in a spectral range of 420-800 nm [32], and then relayed it to the spectral decomposition part. This part consists of a stack of five dichroic long-pass filters with progressively longer cut-off wavelength placed in front of a backing mirror. Each of them reflects a certain wavelength band towards a specific detector or parts of a camera sensor. As a result, signals from each cell are separated into several sub-images: brightfield, side scatter and the set of fluorescent images, corresponding to the different spectral bands. This assembly generates simultaneously six spectral fluorescent images of each cell and follows by the second objective lens, focusing this light at the time delay integration (TDI) camera and forming images in a row on the detector. Latest ImageStream^®^X (Luminex) systems support two cameras simultaneously, providing up to 12 images per object at a 60x magnification, with the diffraction-limited resolution of an image. The lasers are staffed correspondingly to suppress the cross-talk between two cameras. Although, the image resolution in currently available commercial cytometers is diffraction-limited, there are some new developing techniques to overcome this limit [33].

Another optical arrangement is realized in the opensource imaging cytometer SPI [34]. This design makes extensive use of the existing SPIM structured light imaging system [35,36] and adds flow cell with the appropriate fluidic system based on the syringe pump. The optical system, in this case, consists of a laser collimation scheme forming a light-sheet either by a Bessel beam and scanning it with the galvo mirror or by static arrangement where the light-sheet is formed by pair of cylindrical lens and objective lens. Then the usual microscopic tube is placed perpendicular to the light-sheet plane that consists of another objective lens with better resolution/higher numerical aperture, intermediate camera tube lens, and scientific camera.

### 2.3. Camera

Most flow cytometers employ imaging systems based on charged coupled device (CCD) or complementary metal-oxide-semiconductor (CMOS) sensors, which have a number of differences [37]. Additionally, point detectors such as photomultiplier tube (PMT) or avalanche photodiode can be employed. Different sensor types solve the trade-off between temporal resolution and spatial resolution. In comparison to point detector systems, camera-based flow cytometers offer direct imaging of single cells along with the information on morphology and texture at a single-cell level. The main challenge in imaging systems is the minimization of optical blur occurred due to the high rate of cell movement [38].

CCD sensors are typically composed of a two-dimensional array of metal-oxide-semiconductor CCD cells where every diode provides a charge build-up during image capture. Each photodiode serves as a discrete sensor that corresponds to the pixels of the final image. Therefore, the size of the imaged object is defined by the size of the photodiode. The CCD cameras are limited by the sequential transfer time, which determines the rate of the image acquisition as all pixel charges must be transferred before a new image can be captured. In a number of commercial imaging flow cytometers (such as Amnis^®^ (Luminex), IN Cell Analyzers (GE Healthcare Life Sciences, Little Chalfont, UK), ImageXpress^®^ (Molecular Devices, San Jose, CA, USA) CCD sensors accompanied by wide-field illumination. Alternatively, ImageStream^®^ (Luminex) and FlowSight^®^ (Luminex) cytometers employ high-speed time delay integration CCD cameras [39,40]. This type of CCD sensor was designed to image fluorescent objects moving along one axis, which is provided by using multiple rows of sensors pixels that shift charge synchronously with the movement of an object along the same axis. This approach allows increasing the signal from each object by several orders of magnitude (from the microsecond scale to the millisecond scale) without an increase of noise, thus leading to high sensitivity of the system.

Similar to CCD, CMOS sensors also employ a two-dimensional array of photodetectors to perform photoelectric conversion of detected signals. The main difference comparing with CCD is CMOS sensors amplify the signal at each sensor, followed by the storage of the overall information. This approach significantly reduces the time of single image acquisition. Additionally, each pixel acts as an individual amplifier increasing the pixel-to-pixel variability. Further, in 2016 Fairchild Imaging improved the design of the CMOS sensors resulting in low read noise and high quantum efficiency (up to 82%). The new design was named scientific CMOS (sCMOS) [41].

The choice between CCD and sCMOS cameras depends on the particular task and should take into account the pixel size, frame rate, quantum efficiency, spectral response, dynamic range, and noise of the sensor. In general, sCMOS cameras are characterized by higher frame rate, enhanced dynamic range, and larger field of view (up to 4–16 times larger area), while CCD provides higher sensitivity and uniformity over the sensor, which is important for super-resolution and multiphoton applications [42]. Modern electron-multiplying ССD (EM-CCD) sensors have a better low-light sensitivity and increased dynamic range in comparison to CCD, but they have a larger pixel size [43]. The sCMOS sensors also offer an option to improve imaging quality. For instance, the sCMOS sensor can operate in a global shutter mode (all pixels are measured simultaneously) and rolling shutter mode (sensor is read line by line and an only single line is active) [44]. Using a rolling shutter can improve contrast and increase image quality, however, it can suffer from image artifacts when measuring high-speed moving objects.

Both CCD and CMOS detectors may be combined with light-sheet illumination to employ optical sectioning to reduce photo-induced stress of the biological samples. The light-sheet approach uses illumination made perpendicularly to the axis of observation [45]. The excited fluorescence from the light sheet is then projected onto an imaging sensor [46]. The main advantage of light-sheet is the relatively low irradiation of fluorescent molecules during the image acquisition in combination with the good resolution and image contrast that makes it very popular in imaging in vivo, measurement of rapidly moving and changing objects with the minimum damage [47,48].

On the other hand, the employment of camera-based systems may be limited by the weak fluorescent signal and the high speed of cell flow. To overcome this, a number of PMT-based detectors are employed. PMTs are equipped with a light-sensitive photocathode that converts light into photoelectrons, which are further amplified by acceleration by dynodes voltage [49,50]. Although the quantum efficiency of PMTs is not high (about 20%), they provide sufficient amplification (up to thousands of times) of the detected signal. To date, multiple discrete PMT detectors are employed in most of confocal laser scanning microscopes (CLSM) to detect fluorescent signal from multiple fluorescent probes (up to five simultaneously) [51] and in imaging flow cytometers, such as Acumen^®^ (LabTech, Melbourn, UK), iGeneration instruments (CompuCyte, Westwood, MA, USA), and iCyte^®^ (ThorLabs, Newton, NJ, USA). These systems employ optical time-stretch imaging in order to capture fast objects in real-time at the rate of tens of MHz. PMTs offer higher sensitivity and higher dynamic range due to electron multiplication and gain adjustment within the sensor. PMT-based flow cytometers also demonstrate higher bandwidth with reduced noise in comparison to CCD and sCMOS sensors, which helps to increase throughput. However, PMT does not provide spatial information, which limits the number of its applications. Some systems combine PMTs with high-speed microscopy laser scanning techniques in order to provide full information from the sample. For instance, Cellomics^®^ ArrayScan^®^ VTI (Thermo Scientific, Waltham, MA, USA), and Operetta^®^ (Perkin Elmer, Waltham, MA, USA) commercial flow cytometers employ CLSM schemes. In conventional CLSM, single-pixel detectors or PMTs are employed to collect the signal. A specimen is scanned pixel by pixel over the surface by the focused spot to build an entire image [51]. Additionally, the so-called multi-field of view imaging flow cytometer (MIFC) method can be applied to provide imaging of multiple channels simultaneously, which significantly increases throughput [52].

Mckenna et al. developed a microfluidic flow cytometer containing 384 parallel channels with one-dimensional multicolor confocal fluorescence imaging to provide a collection of information from rare-cell samples [53]. A combination of point detector with an optoelectronic image-encoding/decoding time-stretching method allows for reaching high image quality and increased throughput [54]. Mapping images into the radio frequency spectrum using conventional cameras provided increased readout methods comparing to usual methods [55]. The flow cytometry system based on precisely-defined focal spot and a line scan CMOS sensor was first developed by Heng in 2011 and allowed the parallel collection of multiple fluorescence channels with high resolution but low throughput (1000 cells/s) [56].

## 3. Cell Labeling

The most straightforward and reasonable approach to distinguish one type of cell from another one is the direct labeling of cells with an appropriate and well detectable label. At the moment, there is a number of methods used for cell labeling, based on which the separation of target cells from the bulk can be performed. Either fluorescent, magnetic, or acoustic properties of various tags are successfully applied for cell sorting. This section highlights the basic currently used cell markers.

### 3.1. Fluorescent Labeling

Fluorescent labeling is probably the most common and well-developed technique to distinguish one type of cell from others. Currently, a great number of fluorophores with different spectral properties are widely used in microbiology, cell biology, and medicine. Modern fluorescent dyes are used to label specific structures on the surface and inside the cell and to estimate the rate of metabolic processes. In addition, vector-encoded fluorescent proteins are also used for cell labeling.

#### 3.1.1. Fluorescent Label Conjugated Antibodies

Attachment of fluorescent labels to cell populations of interest via antibodies is widely employed in cytometry and cell imaging analysis. Generally, antibodies are immunoglobulins with a similar structure (Figure 2) including heavy (H) and light (L) chains linked to each other by disulfide bonds and are composed of the variable (V) and constant (C) regions [57,58]. Polymeric immunoglobulins have an additional J chain responsible for polymerization [57,59]. Antibodies are able to bind with a variety of antigens due to V regions, which is also known as fragment antibody binding (Fab) regions, at the ends of H and L chains [57,58]. Antigen-antibody interactions depend on the three-dimensional structure of the Fab region determined by the amino acid sequence and their charge [60]. The affinity of antigen-antibody binding is driven by entropic (hydrophobic) and enthalpic (Van der Waals, hydrogen bonding, salt bridge) effects, or by entropy-enthalpy compensation [61]. Additionally, to describe the mechanisms of antigen-antibody interaction two main theories (“Key-Lock” and “Induced Fit”) based on a conformational change of proteins were derived [61,62].

Thus, the strategy of cell labeling is based on the fact that specific cell populations express defined surface markers that are unambiguously identified by the corresponding antibodies conjugated with a fluorescent label. The main benefit of this approach is the possibility of direct cell labeling in vivo by the injection to the blood circulation [63,64,65]. A variety of fluorescent labels to conjugate with antibodies is available, including small-molecule organic dyes, quantum dots, fluorescent proteins, polymer dyes, and tandem polymer dyes.

Small-molecule organic dyes are the simplest and most reasonable choice to conjugate to antibodies. They are commonly used due to relatively good photostability and ease of preparation. Indeed, the protein conjugation protocols are well established, and there are commercially available conjugation kits allowing to attach the most common dyes to the desired antibody as well as already conjugated antibodies that may be purchased from suppliers. There are several widely used small-molecule organic dyes. Among them, probably the most popular are derivatives of fluorescein, rhodamine, and cyanine dyes.

Fluorescein and its derivatives exhibit fluorescence due to a multi-ring π-conjugated aromatic xanthene structure [66]. The absorption maximum of the dye is in the region of 488–495 nm and, thus, can be effectively excited by the argon 488 nm laser. The maximum of emission spectra is typically between 512 and 525 nm with a relatively high quantum yield [67,68]. The fluorescein derivatives are usually prepared by the substitution of either fifth or sixth carbon of the lowest ring. Fluorescein-5-isothiocyanate (FITC) is the most popular fluorescent label ever created. The conjugation of a protein molecule with the FITC label is driven by the formation of isothiourea linkage between the protein primary amine group and the FITC isothiocyanate group [69]. Recently, Chaganti et al. described the modified protein FITC labeling technique employing tandem affinity purification tags at the N- and C-termini of the target protein [70]. Currently, FITC is widely used for labeling various entities of biological nature, such as peptides [71,72], antibodies [73,74] and, polysaccharides [75].

Rhodamine structure is similar to that of fluorescein except for the oxygen atoms in the outer rings are substituted by the nitrogen. All rhodamine-based labels are derivatives of this primary structure. The most widely used modification for biomolecule labeling and further fluorescent imaging is tetramethylrhodamine-5-isothiocyanate (TRITC) [76]. Comparing to the initial rhodamine molecule, TRITC has two methyl groups attached to each nitrogen atom on the outer rings and fifth carbon atom at the lowest ring-substituted to isothiocyanate group. The TRITC molecule has the spectral maximum of absorbance at 545 nm, and the emission maximum at 570 nm. The mechanism of TRITC conjugation with protein molecules is the same as for FITC due to the isothiocyanate functional group [69]. In comparison with FITC, TRITC has a smaller quantum yield, however, it is more photostable, and its fluorescence is less affected by an external medium. Additionally, the TRITC emission band is better for fluorescent imaging in vivo or in biofluids. Texas Red (sulforhodamine 101) is another rhodamine derivative widely used for fluorescent protein labeling [77]. It contains sulfonate groups in a third and fifth position of the lowest ring that can form stable sulfonamide bonds. Texas Red has the maximum absorption at 589 nm and the maximum of emission at 615 nm, along with a relatively high quantum yield. Since the maximum of absorption and emission are shifted to the red region, Texas Red perfectly suits the dual-mode fluorescent imaging combining with fluorescein dyes.

Conventional cyanine dyes generally consist of two cationic ring structures, including nitrogen atoms, and one of them must carry a positive charge while the rings are linked with an unsaturated polymethine chain [78]. The ring structure may vary from single rings containing five or six atoms in the cycle to complex conjugated structures consisting of several aromatic rings. The length of the polymethine chain also varies and may include from one to seven carbon atoms indicating different subfamilies of cyanine dyes with various spectral properties. Therefore, Cy3 dyes usually absorb and emit light at 500 nm band, whereas Cy5 and Cy7 dyes have maximums of absorption and emission at 600 nm and 700 nm bands respectively. An additional modification of the ring structure allows for fine-tuning of spectral properties by shifting the spectral maximums to the blue or red region. However, the most attractive cyanine dyes are those with the spectral maximum of emission in the near-infrared (NIR) region due to the great suitability for bioimaging applications. Thus, the structure of the dyes keeps being optimized to design the dyes with the enhanced properties. For instance, Li et al. designed a NIR frequency upconversion Cy7-NO2 probe for the detection of a nitroreductase enzyme based on Cy7 dye [79]. Ge and Tian reported on the fluorescent probe based on cyanine dye coupled with Zn-Dpa recognition units to monitor p-tau protein in a single neuron [80]. In order to make cyanine dyes suitable for proteins conjugation, the sulfonated groups are introduced to their structure. Generally, commercially available cyanine dyes have from two to four sulfonated groups for the optimal conjugation in aqueous solutions.

Quantum dots (QDs) are nanosized semiconductor crystals that emit photoluminescent light due to electron-hole recombination [81]. QDs have several advantages comparing to conventional small-molecule fluorophores. First, QDs combine high quantum yield (0.1–0.9) with a large molar extinction coefficient (10^5^–10^7^ M^−1^cm^−1^) that results in a bright luminescence [82]. QDs have a broad absorption band and narrow symmetric fluorescence spectra (full width at a half-maximum of about 25–35 nm) shifted from the absorption peak to 100 nm [82]. This allows to avoid spectral overlapping during multiplex detection experiments and thus less compensation is required between QD channels. The spectral maximum (“color”) of emitted light may be adjusted by the size and material of QD crystals [83]. Finally, QDs are much more photostable and have longer lifetimes of the excited state that makes them more suitable for dynamics tracking experiments [84]. Typically, QDs are referred to as I−VI, III−V, and IV−VI binary compounds and their core/shell structures [85]; however, these compositions have biocompatibility issues. With this regard, in recent years, the QDs prepared of carbon materials attract much attention [86,87].

However, the main difference between QDs and organic dyes is in protein conjugation. Small-molecule fluorophores typically have single reactive group coupling with specific sites on a protein molecule [88]. The general principle is to mix the excess of the dye with the target molecules. As the size of fluorescent labels is comparable with the size of amino acid residues, the target protein structure generally remained the same, preserving the specificity and avidity of the antibody. Unlike the small-molecule dyes, QDs have a much larger size and a considerable surface area that can be modified by the biomolecules at many different sites as well as with irregular orientation of the antibodies on the QD surface, which affects biological activity [89]. This makes challenging the conjugation of QDs with antibodies regarding effective cell labeling and sorting. However, there are two strategies of QD controllable conjugation. These are covalent conjugation methods and self-assembly based techniques. The covalent conjugation implies the formation of covalent bonds between the target antibody and QD surface ligands. Although conventional covalent conjugation does not allow for the control over the number of conjugated proteins and their orientation, the recent advances in developing of chemoselective ligation [90,91] and bio-orthogonal [92] reactions have resulted in effective and controllable conjugation of QDs and biomolecules [93,94]. On the other hand, self-assembly based conjugation employs specific high-affinity interactions without the formation of covalent bonds. The typical example of this specific recognition is tightly binding between biotin (vitamin B7) and avidin-like proteins that are known as the strongest non-covalent bond in nature [95]. The basic strategy for conjugation is a modification of the QD surface with avidin-family protein that binds with a biotinylated molecule of interest [96]. To date, there are commercially available kits to attach biotin to any biomolecule as well as QDs modified with streptavidin for the following specific binding. However, conjugation via biotin-avidin binding is limited in control over conjugation valence (i.e., the number of molecules attached to a QD) [97] and orientation of antibodies. Currently, one of the emerging self-assembly strategies for QD bioconjugation providing the best overall control is the exploiting of polyhistidine tag [98]. Polyhistidine is a sequence in a protein molecule consisting of six histidine residues with an affinity for divalent metal cations. Polyhistidine tag is typically bonded with a molecule of interest and then coupled to metal atoms on the QD surface. The advantage of polyhistidine self-assembly is the fact that it does not require additional QD surface modification, employs well-established protocols and provides reasonable control over resulted stoichiometry [99]. Currently, the preparation of more sophisticated polyhistidine tags is carried out providing less steric hindrance during assembly with QDs and better binding affinity with biomolecules [100].

Fluorescent phycobiliproteins were considered as probes for immunofluorescent experiments since the 1980s [101]. These naturally arisen fluorophores are a part of the photosynthetic complex of certain cyanobacteria and algae. The main function of phycobiliproteins is to absorb and transfer the light energy to adjacent chlorophyll molecules for further utilization in the cell life cycle. Thus, they are naturally designed to have a large quantum yield (up to 0.98) and molar extinction coefficient (2.4 × 10^6^ M^−1^cm^−1^ at 545 nm for B-Phycoerythrin) that are far beyond of those of small-molecule dyes [69]. Typically, phycobiliproteins contain multiple chromophoric bilin groups that are linear tetrapyrroles covalently bonded to apoprotein (Figure 3).

Depending on the number and organization of bilin groups, there are four main types of pigments in phycobiliprotein structure: yellow-colored phycourobilin, red-colored phycoerythrobilin, blue-colored phycocyanobilin, and purple phycobiliviolin [102]. The resulted protein spectral properties are defined by the relative content of each pigment in the protein structure [103]. For instance, the two main forms of phycoerythrin employed as fluorescent labels, which are B-Phycoerythrin and R-Phycoerythrin, contain 34 bilin groups and have a broad absorption band from green to yellow region of the visible spectrum with multiple peaks and maximum of emission at 575–578 nm. This makes them perfect for bioimaging as their fluorescent signal is detected in a red and near-infrared region along with variable excitation sources. Phycobiliproteins can be conjugated with antibodies, avidin, biotin, and other biomolecules [104,105,106] without losing their fluorescent properties via common heterofunctional crosslinkers, including N-succinimidyl 3-(2-pyridyldithio)propionate (SPDP), succinimidyl-4-(N-maleimidomethyl)cyclohexane-1-carboxylate (SMCC), and succinimidyl-4-(p-maleimidophenyl)butyrate (SMPB) [69]. The crosslinkers react with amine groups of phycobiliproteins producing activated sites for the coupling of sulfhydryl-containing molecules. The targeted biomolecule can be modified with thiol groups to complete the conjugation. However, despite their outstanding brightness, one should consider that phycobiliproteins are susceptible to photobleaching and are not well suitable for long-term dynamics monitoring.

Recent advances in polymer chemistry and the development of organic optoelectronic systems gave rise probably for the most sophisticated type of fluorescent labels that are engineered polymer dyes. Analogous to phycobiliproteins, polymer dyes consist of a number of optical subunits that are able collectively to absorb light and respond to quenching. Structurally, polymer dyes are specially designed π-conjugated polymers with multiple π-orbitals that can absorb and emit light due to electron delocalization similar to small-molecule organic dyes [107]. However, unlike small-molecule dyes, the absorption and emission of light by polymer dye molecule co-occurs in many sites over the polymer chain. This multiple impact results in relatively high quantum yield (up to 0.65) and a molar extinction coefficient comparable to that of QDs and phycobiliproteins (about 10^6^ M^−1^cm^−1^). Furthermore, the length and structure of the polymer chain can be modified to adjust the spectral properties of the resulted dye. Additionally, polymer dyes are designed to be very photostable and absorb light only of a specific wavelength (i.e., to have a narrow excitation band) that makes them suitable for multiplex and long-term detection experiments [108]. Polymer dyes can be conjugated with antibodies via conventional conjugation protocols as they are pre-synthesized with defined specific binding groups providing well-controllable coupling considering both the number of conjugated labels and location of binding sites. The example of polymer dye is Brilliant Violet BV421 fluorochrome.

Additionally, phycobiliproteins and polymer dyes can be modified with other fluorophores to create dyes with modulated spectral properties employing fluorescence resonance energy transfer (FRET). The fluorophores are combined to form donor-acceptor couples so that the emission band of one fluorophore overlaps with the absorption band of another one. This results in extremely bright tandem dye structures with enhanced Stokes shift up 300 nm that can be effectively excited by two laser sources. Formation of tandem dye structures may be employed to shift the emission of the initial fluorophore to the red or near-infrared band, which is beneficial for bioimaging applications, or in case of low antigen density. However, one should consider that tandem dyes are quite unstable and have rather variable energy transfer efficiency. The examples of tandem dyes are the conjugates of B-Phycoerythrin and Allophycocyanin phycobiliproteins; R-Phycoerythrin and Cy-5 dye; B-Phycoerythrin and Texas Red dye; the family of Brilliant Violet dyes based on BV 421 polymer core.

#### 3.1.2. Cell Tracking Dyes

Fluorescent cell labeling can be performed by using fluorescent cell tracking dyes. These dyes allow for analyzing cell cycle, migration, proliferation, permeabilization of the membrane, etc. by staining different components of the cell. Furthermore, the dyes being incorporated into the live cells allow for their distinguishing without affecting cell metabolic activity. Coupling of tracking dyes with antibodies provides the additional possibility to carry out a phenotype analysis due to the formation of the dye-antibody combination. Depending on the affinity, fluorescent dyes are divided into the three main groups that are nucleic acid, cytoplasmic, and membrane-bound dyes.

Nucleic acid dyes typically bind with DNA and/or RNA. They are generally used to quantify DNA and to observe cells distribution during the cell cycle (Propidium Iodide, 7-Aminoactinomycin D or 7-AAD, DyeCycle Violet, 4′,6-Diamidino-2-phenylindole or DAPI, Hoescht 33342), to estimate cell viability (PI, DAPI), to isolate chromosomes for sorting (Hoescht 33342, Chromomycin A3), to quantify apoptotic cell fractional DNA content (Acridine Orange or AO, 7-AAD), etc. [109,110] These dyes are frequently used for cell migration and tracking analysis due to their high retention. Nevertheless, DNA-binding dyes are not applicable for proliferation analysis because the content (and intensity) of the dye decreases after each cell cycle [111].

A large family of fluorochromes developed for bioimaging is prepared from “acridine derivatives”. The chemical structure of fluorochromes from the acridine family is performed by several aromatic rings forming xanthene that produces a planar configuration responsible for the binding to the specific target. Physico-chemical behavior is driven by the three-dimensional conformation of the target site. Other interactions, such as hydrogen and Van der Waals bonds, can enhance or reduce binding mode depending on the reaction conditions. The most widespread and frequently used fluorochrome is Acridine Orange (AO). This dye is appreciated due to its metachromatic nature: according to the microenvironment, AO can vary the emission spectrum. It emits green fluorescence when bound to double-stranded DNA and red fluorescence when bound to single-stranded DNA or RNA. The excitation/emission maximum wavelengths are 460/650 nm for RNA-binding and 500/526 nm for DNA-binding analysis. This fascinating property provides its usage for differentiation of double- and single-stranded configurations that are in detail described in the literature devoted to cell kinetics, damage, and apoptosis. The AO dye can be sufficiently excited by a 488 nm blue line of an argon laser.

7-Aminoactinomycin D or 7-AAD is the fluorochrome that belongs to the group of dyes forming a complex with a guanine base (in the presence of magnesium ions). It selectively binds to guanine-cytosine regions of DNA and provides an explicit banding pattern in polytene chromosomes and chromatin for chromosome banding studies. This family also includes mithramycin, chromomycin, and olivomycin dyes with similar chemical behaviors. The absorption maximum is 546 nm, but 7-AAD is effectively visualized by a 488 nm flow cytometer laser line. This fluorescent intercalator performs spectral shift upon binding with DNA and emits light in the far-red region, having an emission maximum at 647 nm. It makes nucleic acid stain useful for multicolor fluorescence microscopy and flow cytometry. 7-AAD is used for cell cycle analysis in bioimaging, but it encounters some obstacles during its utilization. As well as it cannot penetrate the intact cell membrane, it is generally excluded from live cells and proposed for apoptotic cell death study. Nevertheless, in the case of fixed and permeabilized cells, the 7-AAD is applicable.

Another group of DNA-binding fluorochromes is presented by diamino-phenylindole derivatives; one of those is 4′,6-diamidino-2-phenylindole (DAPI). DAPI binds to adenine-thymine regions of DNA. This nuclear counterstain has the absorption maximum in the ultraviolet (UV) region and provides blue-fluorescent emission at 470 nm. Although the spectral maximum of absorption is at 350 nm, DAPI can be excited by a 405 nm laser line while exhibits high quantum yield. The spectral properties make DAPI applicable for multicolor flow cytometry experiments, especially in combination with green (FITC, GFP) and red (Rhodamine, Texas Red) fluorophores. In bioimaging, DAPI is in common use for counting cells, estimation of apoptosis and cell viability, sorting cells based on DNA content due to high affinity for DNA, and as nuclear segmentation tool in imaging analysis. Generally, this fluorochrome can stain fixed cells because of its impermanence, but at higher concentrations, it is possible to use DAPI for live cells. However, one should take into account that DAPI has a wide emission range (96 nm at a half-maximum) and may bind non-specifically when the staining period is longer than in the approved protocol.

One more popular UV-excited live cell stain is Hoechst 33342. The dye can permeate the intact cellular membrane and bind to DNA even of live cells without their further damaging. Thus, it is employed for cell cycle analysis analogous to DAPI. The mechanism of staining is similar to DAPI, as well as the excitation/emission wavelengths (361/497 nm, respectively).

Propidium Iodide (PI) and Ethidium Bromide (EB) are the two main dyes of intercalating phenanthridine dye family to perform analysis of proliferation and cell viability. As they are not able to permeate the intact membrane of live cells, they are frequently used to differentiate dead cells in a population. Thus, staining with these dyes requires to fix the cells or to permeabilize them for further DNA binding. As an acridine derivative, PI can bind both with double-stranded DNA and with double-stranded RNA chains due to interaction with amines within a cell to determine if the cell membrane is intact. The EB reacts in the same way. Furthermore, PI and EB have a specific spectral feature: they are almost non-fluorescent as free molecules in aqueous solution but provide highly increased quantum efficiency when intercalated into nucleic acids. The free molecules excited by the adequate light tend to dissipate the energy to the surrounding polar water molecules. In turn, the intercalated molecules can only disperse the energy by the light emission. The excitation maximum of PI is 535 nm, which is appropriate for the 488 nm argon laser line, with the spectral maximum of emission at 617 nm. Although the large variety of new DNA-affinitive dyes that have been developed in the last years, PI is still the most commonly used for cell cycle analysis and DNA/RNA detection.

Cytoplasmic dyes have been widely used for cell migration and proliferation studies due to their longer retention in comparison with nucleic acid binding dyes [112,113,114,115]. However, they exhibit a higher level of cytotoxicity to cells as they bind to cytoplasmic proteins and therefore affect the cellular enzyme functions [116]. The most prevalent cytoplasmic dyes are fluorescein-based carboxyfluorescein succinimidyl ester (CFSE), calcein acetoxymethyl (calcein-AM), calcein violet acetoxymethyl (calcein violet AM), and resazurin-based alamarBlue, and CellTiter-Blue.

The dyes of the fluorescein group can easily penetrate the intact membrane of live cells due to their hydrophobic properties. Afterward, the dye molecules are hydrolyzed by intracellular esterases that leads to sufficient enhance of the intrinsic fluorescence. Concerning calcein AM, after its transport into the cell, the esterases remove two acetoxymethyl groups, and non-fluorescent molecule converts to a green-fluorescent calcein. The intensity of the resulted overall fluorescent signal depends on the esterase activity that is directly proportional to the number of viable cells. In turn, the dead cells lack the active esterases that prevent a calcein-AM conversion to the fluorescent form. The spectral maximum of calcein extinction is at 495 nm, and therefore, this dye can be effectively excited by a flow cytometry setup equipped with 488 nm argon laser. The spectral maximum of emission corresponds to 515 nm. Generally, calcein is exploited for a variety of studies, including cell adhesion, chemotaxis, drug resistance, cell viability, apoptosis, and cytotoxicity.

Resazurin-based dyes, for example, AlamarBlue, are the type of cytoplasmic fluorochromes, in which fluorescent properties are caused by the reaction of chemical reduction. Once trapped by the live cell, resazurin reduces by mitochondria and converts to the fluorescent form called resorufin. Thus, the AlamarBlue dye has a 560 nm emission wavelength maximum with another band at 590 nm in the red region. High fluorescence signal allows us to use it in flow cytometry studies for cell viability, cytotoxicity and proliferation analysis. However, it should be noted that AlamarBlue fluorescence may overlap with the fluorescence regions of other compounds being tested. Additionally, it can provide a minor toxic effect on the stained cells.

Membrane-inserting fluorochromes are represented by lipophilic dyes (dialkylcarbocyanine derivatives) that diffuse laterally within the cellular plasma membrane resulting in the entire cell staining at dye optimal concentrations. Dye molecules localize at the membrane surface anchoring to it by lipophilic “tails”. Carbocyanines have short alkyl tails that attached to the imine nitrogens; thus, they can be used both as membrane-potential sensors and as organelle stains for mitochondria and the endoplasmic reticulum. Those with longer alkyl tails provide long-term labeling of cells. In addition, this type of dye is likely to be less toxic than the previous (nucleic acid and cytoplasm-binding) as well as it possesses longer retention [117].

The main dialkylcarbocyanine dyes are 1,1′-dioctadecyl-3,3,3′,3′- tetramethylindodicarbocyanine, 4-chlorobenzenesulfonate salt (DiD), 1,1′-dioctadecyl-3,3,3′,3′-tetramethylindotricarbocyanine iodide (DiR), 3-octadecyl-2-[3-(3-octadecyl-2(3H)-benzoxazolylidene)-1-propenyl]-, perchlorate (DiO,) and 1,1’-dioctadecyl-3,3,3’3’-tetramethylindocarbocyanine perchlorate (DiI). The fluorescence of these dyes is greatly enhanced after incorporation into membranes or binding to lipophilic biomolecules weakly fluorescent in water, such as proteins. Along with it, dialkylcarbocyanine derivatives perform high molar extinction coefficients, polarity-dependent fluorescence, and short excited-state lifetime (~1 nanosecond). This is an advantage for flow cytometry applications because it allows for more excitation/emission cycles during flow transit. Additionally, dialkylcarbocyanines are quite photostable and provide a suitable imaging tool for flow cytometry study of live cells due to the distinct fluorescence colors. In particular, DiO emits green light (abs. 484 nm/em. 501 nm), DiI emits orange light (abs. 549 nm/em. 565 nm), DiD emits red light (abs. 644 nm/em. 663 nm), and finally, DiR emits deep red light (abs. 748 nm/em. 780 nm). Among them, DiO can be effectively excited by 488 nm argon laser, and its fluorescence spectrum is comparable with that of FITC, whereas the 633 nm He-Ne laser is convenient for DiD. TRICT and DiI can be visualized by the same optical filter sets due to their excitation/emission wavelengths similarity. Lipophilic carbocyanines as membrane-bound counterstains have been appreciated for their use to visualize membrane fusion and cell permeabilization, for cell migration and proliferation studies [116,118].

#### 3.1.3. Fluorescent Proteins

One more approach to distinguish cells of interest is to make the cells fluorescent themselves without external labeling that might potentially affect their behavior. This is achievable through the modern molecular genetic methods allowing for the transfer of fluorescent protein (FP) genes into the cells. The FP labeling for bioimaging is one of the most widespread applications of FPs. The general strategy for cell labeling is to include the nucleotide sequence of FP responsible for fluorescence in the gene tailor the gene body of the labeled cell protein depending on its structure. In some cases, a flexible Gly-rich linker between the FP and the labeled protein is used in order to prevent potential steric conflicts [119]. The most significant advantage of FPs is the possibility to monitor the dynamics of the desired cell population in extremely long-term experiments as all next cell generations preserve bright fluorescence, even after multiple divisions. Additionally, FPs are suitable to study the mobility of proteins inside a cell [120], for visualization of intracellular structures [121], and as markers to highlight a promoter activity in gene engineering and cell biology [122]. Furthermore, the FPs that can shift their spectral maximum of emission in time, thereby allowing them to determine their expression retrospectively, are of particular interest [123]. The FPs with DNA and RNA binding domains are used for real-time labeling and tracking of mRNA [124], DNA, and various structures associated with DNA [125,126,127,128]. The creation of cell lines expressing a certain amount of several fluorescent proteins allows for carrying out multicolor cell labeling with more than 100 shades. This technique is applied in immunology, neurobiology, and transplantology [121].

Currently, several types of FPs are available with different spectral properties determined by variations of their amino acid sequences [129] and, therefore, their structure (Figure 4). Generally, FPs are classified according to their spectral maximum of emission on the blue (BFPs; 440–470 nm), cyan (CFPs; 471–500 nm), green (GFPs; 501–520 nm), yellow (YFPs; 521–550 nm), orange (OFPs; 551–575 nm), red (RFPs: 576–610 nm), and far-red (FRFPs; 611–660 nm) FPs [130]. However, despite the variety of “colors”, the blue, cyan, and yellow emitting FPs are mutated allelic forms of GFP, which was first isolated from the jellyfish *Aequorea Victoria* [131]. GFP contains a fluorophore group that includes the sequence of three amino acids: the serine, tyrosine, and glycine located at 65, 66, and 67 sites. This group forms the imidazoline-5-one heterocyclic nitrogen ring by reacting between carboxyl carbon of serine and the amino nitrogen of glycine and subsequent conjugation with tyrosine [131,132]. The initial GFP originally isolated from the *Aequorea Victoria* has major and minor excitation peaks at 395 and 475 nm with emission peaks at 503 and 508 nm, respectively, and a quantum yield of 0.79 [133]. Additionally, there are several Aequorea protein derivatives with modified properties, such as eGFP [134], Superfolder GFP [135], YFP [136], TagCFP [137]. Furthermore, some GFPs were obtained from other organisms. These are Amazing green (from stony coral, *Galaxeidae*) [138], dendGFP (from octocoral *Dendronephthya sp*.) [139], TurboGFP (from *Pontellina plumata Copepoda*) [140].

RFPs emitting light in yellow, orange, red, and far-red regions were isolated from some species of Anthozoa class. The first RFP was isolated from *Discosoma striata* and called DsRed [142]. The fluorophore of the DsRed protein includes glutamine, tyrosine, and glycine, located at 66, 67, and 68 sites. The RFP has broad excitation and emission spectra with the corresponding maximums at 558 and 583 nm [143]. Variations in the amino acid sequence of RFP, for instance, the mutation of lysine to methionine at location 83, is shifting its spectral maximum of emission to 602 nm [144]. After discovery of DsRed, many derivatives of this protein, such as a DsRed2 [145], DsRed-express [146], DsRed-Monomer [147], mCherry [148] were obtained. In addition, RFP may be obtained from other sources; for example, HcRed1 was derived from the sea anemone *Heteractis crispa* [148].

Recently, a family of photoactivatable fluorescent proteins (PAFP) attracts much attention due to their ability to change the fluorescent properties under treatment by light energy [149,150]. These include groups of reversible and irreversible PAFP [130]. Reversible PAFP can change their fluorescence properties through conversable alterations of their chromophore conformation [151,152]. Proteins of this group were obtained from the corals of the Pectiniidae family [153]. A group of irreversible PAFP is divided into two classes depending on the switching mechanism. The first one is oxidative decarboxylation that is the carboxylate group that forms a weakly fluorescent neutral form of the chromophore is removed under the irradiation [154]. Another possible switching mechanism is the β-elimination of the peptide bond between alpha nitrogen and alpha carbon of two amino acids participating in the formation of two different conformations of the fluorophore [130].

### 3.2. Labeling by Magnetic Beads

The application of the external magnetic field is another reliable way of highly selective cell manipulation by the external force [155]. This is achievable since biological materials have a very low magnetic susceptibility, and thus, the cells labeled with magnetic particles or that having the intrinsic magnetic properties can be effectively isolated without the interference with the surrounding medium and objects [156]. Therefore, the essential prerequisite of the magnetic sorting is a high magnetic response of the isolated objects (or its absence in case of the negative sorting). If the cells of interest do not possess the high magnetic susceptibility itself (for instance, as the red blood cells), one should consider to label them with an appropriate magnetic tag.

Generally, depending on the magnetic susceptibility, bulk materials can be classified as diamagnetics, paramagnetics, which have no magnetic order, and magnetically ordered materials like ferromagnetics [157,158]. The diamagnetics do not have magnetic dipoles in the absence of an external magnetic field due to the closed-shell electronic structure of the atoms [159]. These materials possess a very low magnetic susceptibility and a negative magnetization response that means the material magnetization is directed oppositely to the applied magnetic field. Ordinary, all biological and organic materials exhibit diamagnetic behavior. Conversely, the paramagnetic materials contain unpaired electrons in their atom-shell and therefore have randomly oriented magnetic dipoles that can be aligned with magnetic field lines [159]. This results in a positive magnetization response that means their magnetization vector is collinear to that of the applied magnetic field. However, the magnitude of their magnetic susceptibility is still very low [160]. Both paramagnetics and diamagnetics demonstrate the absence of spontaneous magnetization without an external magnetic field. On the other hand, ferromagnetic materials reveal intrinsic magnetization induced by the electronic shell structure of their atoms and the crystal lattice type. In these materials, the short-distance exchange coupling between the electrons prevails over the thermal disorientation that results in the collective interaction of their magnetic moments (spins) leading to the long-range magnetic ordering of the entire material volume at the macroscale. As a result, the ferromagnetics have a very high magnetic susceptibility and a positive magnetization response [160]. However, the intrinsic bulk magnetization substantially increases the magnetostatic energy of the material. In order to minimize the overall internal energy, the magnetic domains are formed. The domains are groups of atoms where the collective magnetization vector is oriented along the easy magnetization axis and specifically polarized relative to the magnetization vector of the other domains. The magnetic domains in the bulk ferromagnetics along with a magnetic crystalline anisotropy are the main origins of the magnetic hysteresis that can be interpreted as a delay in magnetization response when the magnetic field is applied. The delay is related to the additional consumption of magnetic field energy for the movement of domain walls and overcoming the anisotropy barrier to align the magnetization vector along the magnetic field direction. However, the formation of magnetic domains is restricted by the material volume. To put in other words, there is a particular size threshold for a ferromagnetic particle, below which to support a single-domain volume magnetization is more energetically favorable rather than to form the domain walls [161]. This is called a single-domain state. With the further reduction of the single-domain particle size, the magnetic anisotropy energy of the particle becomes comparable with the energy of thermal magnetization fluctuations. Thus, under a certain ambient temperature, the magnetization vector can quickly flip over the magnetic anisotropy barrier separating two equivalent easy directions of magnetization and change its orientation to the opposite one [161]. That results in zero time-averaged or particle ensemble-averaged magnetization in the absence of the applied magnetic field. Controversially, when the magnetic field is turned on, the single-domain particle exhibits a typical paramagnetic behavior. It shows the positive magnetization response without magnetic hysteresis along with the tremendous magnetic susceptibility inherent to ferromagnetics. In this case, the particle is named superparamagnetic. The single-domain transition threshold varies for various materials but typically is about 1–50 nm. This perfectly suits for conjugation with the antibodies and immuno-responsive cell labeling. Thus, the combination of these properties makes the single-domain superparamagnetic nanoparticles the best and reasonable choice for the tags for magnetic cell manipulations.

Generally, there are three main approaches to cell labeling for magnetic separation. The first one is the direct labeling of cells of interest by conjugation of the magnetic particle with the corresponding antibody [162]. Another possible option for magnetically driven cell sorting is related to the different internal absorption capacities of various cells (endocytosis) [163]. This allows for effective separation of monocytes with low absorption capacity from macrophages with high absorption capacity. To some extent, a label-free magnetic separation may be considered as a third labeling approach. The label-free separation is effective to sort the cells with a natural magnetic response from the non-magnetic ones [164].

### 3.3. Labeling by Negative Acoustic Contrast Particles

Cell sorting can be performed as well by standing acoustic waves, produced inside microfluidic devices. Standing acoustic wave is an array of alternate nodes and antinodes of surrounding media pressure, formed in resonance conditions [165]. Depending on the object’s density and compressibility, cells are distributed in the pressure node and antinode regions. In the isotonic solutions, such as a physiological solution and phosphate-buffered saline, cells typically exhibit positive acoustic contrast, which leads to their distribution in pressure nodes (Figure 5a) [166,167]. Some polymeric materials, such as elastomers, oppositely have negative acoustic contrast [168]. This means that objects made from elastomers in the standing wave conditions are placed in pressure antinodes (Figure 5a). Therefore, bio-functionalized particles made from these polymeric compounds (negative acoustic contrast particles—NACPs), can be used for cell sorting by the displacement of positive acoustic contrast cells linked with particles in pressure antinodes (Figure 5b) [166,167]. Russom’s group in 2017 had demonstrated the possibility of colon carcinoma cell separation by microfluidic-based microBubble-Activated Acoustic Cell Sorting (BAACS) method [169].

## 4. The Fluidic System for Sample Preparation, Flow Cytometry Measurement, and Cell Sorting

### 4.1. Sample Enrichment by Target Cells

Diagnostics of diseases by the detection of an embolus in the whole blood still is a difficult task. One of the major challenges is the rarity of untypical objects, which are prognostic or diagnostic factors for some diseases. For example, the presence of circulating tumor cells (CTCs) is a prognostic factor for a number of cancers [10,11,12,13,14,15,16]. However, the detection frequency of these objects in relatively small blood volume is extremely low. For instance, the number of CTCs may vary from 0 to 23.6 cells per 7.5 mL of whole blood sample depending on the cancer type, stage, medication, and operative treatment [170]. Thus, the enrichment of the analyzed blood volume by target cells is a necessary prerequisite for reliable diagnostics. There are a number of methods, allowing for sample enrichment, such as red blood cell lysis, density gradient centrifugation, and cell filtration.

#### 4.1.1. Red Blood Cell Lysis

The red blood cell (RBC) lysis method is one of the commonly used approaches for sample enrichment by target cells [171,172]. This method allows for effective elimination of RBCs, the concentration of which in the blood is 4.5 × 106 cells/µL, thus significantly simplify observing of rear objects in a blood sample. The enrichment is based on soft osmotic lysis of RBC by mixing cells with cold water [173] or specialized RBC lysis buffer [171]. The lysis is performed in soft conditions to prevent significant damage to target cells and is timely stopped by adjusting the salt concentration. However, this method has some limitations. For example, it eliminates only RBCs and does not allow to exclude platelets from the sample. With this regard, the RBC lysis method is often used in combination with other methods to receive the appropriate results.

#### 4.1.2. Density Gradient Centrifugation

Another way is the sample fractionation by the density gradient centrifugation method. This method is based on the distribution of objects with different density in density gradient media. This method is widely used to separate macromolecules, viruses, cell organelles, and different cell subpopulations [174]. There are a number of commercial kits based on organic, inorganic compounds, and nanoscale silica particles. For example, Ficoll™ [175], Histopaque^®^ [176], OncoQuick^®^ [177], Percoll^®^ [178] are successfully used for the isolation of mononuclear and cancer cells from a whole blood sample. The principle of the isolation is shown in Figure 6 as in the case of the OncoQuick separation kit. Additionally, this method is used to select bacterial cells from the sample for bacteremia diagnostics [179]. The employment of several media with different density allows us to fractionate the blood sample accurately and prevent fraction contamination by RBC and platelets.

#### 4.1.3. Cell Filtration

Cell filtration is applicable if there is a significant distinction in the size of the target and bulk cells. A great amount of data shows that CTCs are typically larger than the rest of the blood cells that allows for employing filtration for sample enrichment (Figure 7) [180,181]. The small size emboli, such as bacteria, also can be selected by filtration for bacteremia diagnostics [182]. To perform cell filtration, microfluidic devices are equipped with integrated microfilter [183], membrane microfilter [184] along with other variations of devices [182,185] based on this method are used.

### 4.2. Sample Focusing

To carry out flow cytometry measurements, the cells must be aligned in the focal plane of the optical system of the cytometer. This can be done by hydrodynamic or acoustic focusing. Additionally, focusing prevents capillary blockages. Both acoustic and hydrodynamic focusing is used in commercial flow cytometers (Figure 8), although the latter one is more widespread. Moreover, the flow cytometry systems with visualization option (e.g., Amnis^®^ ImageStream^®^ Mark II system) also use hydrodynamic focusing for fluorescent and brightfield imaging. The laser tweezers technique can be used to manipulate single objects in a flowing stream as well, yet it is not used for focusing in flow cytometry setups.

#### 4.2.1. Hydrodynamic Focusing

For the first time, hydrodynamic focusing was described by Reynolds in 1883 [31]. It implicates the codirectional movement of two liquid streams, one of which is the sample suspension, and the second one is sheath fluid. The velocity or density distinctions between the sample suspension and sheath fluids results in the formation of a two-layer stable flow and alignment of the objects in the middle of the channel [186]. Different configurations of microfluidic devices are used to perform hydrodynamic focusing, however, flow cytometry systems typically use configuration based on the coaxial laminar flow [31] (Figure 8a).

#### 4.2.2. Acoustic Focusing

The acoustic focusing is based on the effect of acoustic pressure described by Kundt and Lehmann in 1874. However, only in 2008, the feasibility of acoustic focusing in flow cytometry was described by Kaduchak et al. [187] and a year later, for the first time, acoustic focusing was introduced in flow cytometry setup Attune^®^ (ThermoFisher Scientific), [188]. Acoustic focusing is based on the redistribution of objects with different density in the nodes and antinodes of the standing acoustic wave (Figure 8b). The method is attractive due to its cost efficiency and applicability for biological objects. To perform acoustic focusing, the length of the acoustic wave should be equal width of microfluidic channel divided by a natural number. Currently, the multinode acoustic focusing is developed for parallel flow cytometry devices that allows for improving throughput comparing to conventional flow cytometers [189].

### 4.3. Sorting

Isolation of specific cell types from bulk heterogeneous biological samples (i.e., blood) is a first and necessary step for many biomedical applications. Effective cell sorting significantly improves the quality of the sample analysis. Conventional cell sorting approaches implies either the separation of antibody conjugated cells or label-free sorting. Separation of antibody conjugated cells generally exploits specific cell labeling with magnetic beads or with fluorescent tags. The first one allows us to sort cells by a magnetic field while the second one assists in fluorescence-activated cell sorting (FACS). On the other hand, label-free sorting allows for cell separation with respect to their size, shape, and morphology through filtration, centrifugation, or sedimentation. However, modern tendencies require fast and automated techniques to analyze a large number of cells. In this perspective, microfluidic methods of cell sorting are expected to provide one with more sophisticated solutions to meet the modern demands in a rapid and reliable analysis.

There are several features of microfluidic systems that are relevant to cell sorting applications (Figure 9). First, narrow microscale channels tend to have a laminar liquid flow that aligns the cells into well-ordered streamlines. Furthermore, microchannels have a flow speed gradient over the cross-section area that allows for employing an additional mechanism of hydrodynamic separation that is not possible on a macro scale. The small size of microfluidic chips enables one to apply locally strong gradients of magnetic, electric, and acoustic fields as well as to combine multiple chips in a single sorting device.

To isolate a particular cell from the carrying fluid flow one should apply a force to this cell. Depending on the applied force nature, several general types of separation mechanisms in microsystems are available: mechanical separation by the direct contact with the structure of the separation system based on the cell size, shape, and morphology; employment of external field gradient (e.g., magnetic, electric, and acoustic field); separation by hydrodynamic force via introduction of secondary fluid flows. On the other hand, with respect to cell modification approaches, one can outline the separation of cells conjugated with fluorescent labels, cells conjugated with beads, and a label-free separation. Additionally, label-free separation methods can be divided into active and passive.

#### 4.3.1. Active Separation Methods

##### Electrokinetic Cell Separation

Electrokinetic cell separation exploits various phenomena that may be activated by an applied electric field (Figure 9a). Typically, electrophoresis, dielectrophoresis, and electroosmosis are employed. Electrophoresis is a movement of a charged particle in a uniform electric field under the Coulomb force. Alternatively, electroosmosis is a movement of a polarizable medium in the electric field due to the collective action of the Coulomb force on its molecules (Figure 10) [190]. This may be used to separate several fluid streams.

Finally, the application of the non-uniform electric field causes the movement of cells due to their polarization (Figure 11), which is called dielectrophoresis (DEP) [191]. If the cells are more polarizable then the medium, they will move to the regions with the highest electric field strength (positive DEP—pDEP) and vice versa, in case the cells are less polarizable, they will move to the regions with the lowest electric field strength (negative DEP–nDEP).

One of the first techniques employing an electric field for cell separation was FACS [26]. At its simplest, the cells of interest are labeled with a fluorescent tag via a complementary antibody. Afterward, the FACS device analyzes the fluorescent signal from the cell streamline. Additionally, FACS devices can analyze the light side- and forward scattered by the cells in order to get the information on their morphology and size. The conventional FACS devices are equipped with a vibration stream channel to form the liquid droplets containing the cells at the outlet and a ring charger to charge the droplets with the cells of interest if the fluorescence was detected. The charged droplets are isolated in a separate streamline by the oppositely charged electrode and collected [192]. However, the necessity to form water droplets is the significant drawback of conventional FACS systems that was successfully overcome by the introduction of microfluidic devices.

At resent years, DEP and its modifications have become the most commonly used method of electrokinetic cell separation in microfluidic platforms [193]. Generally, the DEP force acting on the cell is determined by the cell volume and dielectric properties along with the permittivity of the medium and the strength of the applied field [194]. Thus, the essential benefit of DEP is the possibility of both label-free and immunoaffinity cell isolation depending on their size and dielectric properties regardless of the initial cell surface charge. For instance, in the last few years, various DEP setups were shown to be effective for the isolation of circulating tumor cells (CTCs), which are known to have a neutral charge [195] from the healthy blood cells. Bulfoni et al. demonstrated DEP isolation of epithelial-like and mesenchymal-like breast tumor cells from the blood samples obtained from the clinical patients with a commercially available DEPArray system [196]. Kim et al. used DEP for effective isolation of separate cancer cells with a very high efficiency followed by an intracellular enzymatic assay at the single-cell level [197]. Later, the same group proposed a microfluidic DEP setup that allows for precise localization of rare CTCs in the desired area along with a reduction of the sample volume for further molecular analysis [198]. Alazzam et al. fabricated the DEP device for continuous separation of CTCs, from a heterogeneous mixture of cancer and healthy blood cells employing one-sidewall displacement of the electrodes in the microchannel [199]. The resulted non-uniform electric field acted on the MDA-MB-231 human breast cancer cells with pDEP force whereas the nDEP force acted on the healthy cells leading to almost 100% sorting efficiency (Figure 12a). Li et al. demonstrated continuous high-throughput selective capture of CTCs by DEP at arrays of wireless bipolar electrodes [200]. Noticeable feature of this setup is an easy increase of separation area that can significantly improve the volume rate of processing samples.

Another important DEP application is the isolation of stem cells. It is currently attracting much attention due to its high potential in clinical applications and regenerative medicine. Sun et al. reported on microfluidic DEP device with self-assembled ionic liquid electrodes for continuous cell separation [202]. The device was successfully employed for the separation of human adipose-derived stem cells (ADSC) from MDA-MB-231 human breast cancer cells via pDEP with an accuracy of about 85%. Yoshioka et al. employed pDEP for the isolation of human bone marrow-derived mesenchymal stem cells (UE7T-13) [203]. Moreover, the important result of their study was that pDEP effects on UE7T-13 cell gene expression of the surface differentiation marker due to mechanical deformations caused by separation. Additionally, recent studies demonstrated that DEP could be employed for the separation of heterogeneous populations of stem cells with respect to their biological fate. For instance, neural stem and progenitor cell (NSPCs) heterogeneous populations contain both neuron and astrocyte progenitors that give rise to neurons and astrocytes respectively. The separation of neuron and astrocyte progenitors is crucial to improve the purity of transplanted stem cells in clinical therapy. Adams et al. demonstrated that these subpopulations could be effectively separated by the alternate current (AC) DEP due to the different electrophysiological properties of the cell membranes (whole-cell membrane capacitance) revealing at a certain stage of the cell differentiation [204]. Previously, El-Badawy et al. examined the electrophysiological properties of adipose stem cells (ASCs) and bone-marrow mesenchymal stromal cells (BM-MSCs) with AC DEP [205]. They found out the differences in cell traveling and rotation speed that potentially may be exploited for cell separation.

The realization of conventional DEP implies a fabrication of metal electrodes in the microfluidic channel that is potentially harmful to the contacting cells. This leads to the development of DEP modifications to avoid the direct contact of the analyzed cells with the metal electrode surface. For instance, in insulator-based DEP (iDEP), the DEP force originates due to the polarization of dielectric structures placed into the microfluidic channel by applying an external electric field [206]. The metal electrodes are typically placed in the inlet and outlet ports of the device and do not contact with the microchannel interior. First, iDEP was shown to be effective in the separation of live and dead bacteria *E. coli* [207] and later on the separation of different types of live bacteria [208]. As a more recent example, Lewpiriyawong et al. demonstrated the iDEP setup run by the DC-biased AC electric field [209]. The AC controls iDEP force for the cell isolation whereas the DC field generates electroosmotic motion of the liquid through the channel. The combination of AC and DC electric fields was shown to be effective for the cell capturing at the liquid flow rate up 5 mL/min.

Further development of DEP approaches reveals the contactless dielectrophoresis (cDEP) technique. The main idea of cDEP is an exploiting of microchannels filled with high-conductive liquids as so-called liquid electrodes (see Figure 12b). Generally, the liquid electrodes are separated from the main sample-carrying microchannel with an insulating membrane minimizing the contact between the electrodes and analyzing cells [210]. This results in a significant reduction of contamination of biologicals samples, electro-induced heating, electrochemical effects, and bubbles formation. The inhomogeneity of the intrinsic electric field required for DEP separation is achieved through the corresponding geometry of the side liquid-electrode channels and placing of the dielectric elements inside the sample-channel. Shafiee et al. first reported on a successful selective separation of live human leukemia cells from dead ones by cDEP basing on their dielectric properties [201]. Later, Henslee et al. first demonstrated the ability of cDEP isolation of the target MDA-MB-231 human breast cancer cells from the heterogeneous cell mixture [211]. The same group employed cDEP to analyze the area-specific membrane capacitance of red blood cells, macrophages, breast cancer, and leukemia cells [212], which is an important prerequisite for the AC electrokinetic separation. More recently, Hanson and Vargis described the cDEP setup combined with Raman spectrometer for cell isolation and identification on a single device [213]. Rahmani et al. proposed a flow microfluidic cDEP device for continuous bioparticles enrichment employing platinum electrodes separated from the sample-channel with PDMS layer [214]. The employment of metal electrodes instead of liquid ones allows for effective separation with much lower operating voltage due to higher conductivity of the metal compared to the ionic liquid.

However, the production of micro-scale electrodes for microfluidic systems remains quite challenging and requires delicate and complex fabrication processes. To overcome this issue, an optically induced DEP (oDEP) approach was introduced. In oDEP devices, a photoconductive layer on a plate electrode is used to induce the non-uniform electric field inside the microchannel. When the photoconductive layer is illuminated with light passed through a defined pattern, the light-exposed areas become much conductive than the dark areas and therefore form “virtual electrodes” inducing the non-uniform electric field [215]. This allows for avoiding the fabrication of patterned microelectrode structures and makes the separation more adjustable. Chiou et al. first demonstrated the potential of oDEP to be exploited for cell manipulation with the technique permitting a high-resolution patterning of electric fields on the photoconductive surface [216]. The described device enabled a simultaneous manipulation of 15,000 particle traps with an optical intensity 105 times less than that of optical tweezers. Later on, Hwang et al. described the oDEP based technique to discriminate normal and starved abnormal oocytes [217]. In further oDEP development, Huang et al. first characterized the operating conditions for the manipulation of CTCs of prostate cancer (PC-3) and human oral cancer (OEC-M1) and leukocytes with a minor cell aggregation [218]. More recently, Chiu et al. described the oDEP microfluidic system using T-shape microchannel for CTCs isolation and operating with a light bar to manipulate with leukocytes and light circles to capture and isolate PC-3 cancer cells [219]. Further, the same group proposed four-the cascade oDEP CTCs isolation scheme employed after the conventional CTCs manipulation process [220]. This allows for applying up to four isolation conditions simultaneously and implementing a higher-resolution separation. As the most recent example, these authors proposed the microfluidic oDEP device for high-purity isolation of CTC clusters based on their size and preserving clusters integrity for further analysis [221]. Furthermore, oDEP was shown as a versatile tool to determine the cell electrophysiological parameters. They are not only relevant biomarkers to characterize cellular phenotype and state but also the important prerequisites for cell sorting. Liang et al. described a theoretical background to figure out the cell membrane capacitance and conductance with a shell-core polarization model [222]. Afterward, they studied the response of four different cell types to positive and negative oDEP forces with a motion-tracking technique in liquid media of various conductivity. This non-invasive approach is advantageous to get statistically important data from the individual cells to contribute a number of biomedical applications including stem cell differentiation.

Additionally, DEP may be successfully combined with other separation approaches like inertial force microfluidic separation [223], fluid flow fractionation [224], and field-modulated electroosmotic flow separation [225].

As a final remark of this section, we refer to some recent reviews devoted to electrokinetic cell manipulations in microfluidic devices. Adekanmbi and Srivastava have given a deep insight review of DEP manipulations on diseased cells [226]. This includes an overview of DEP techniques and devices, the theoretical background of DEP cell separation, and DEP applications for isolation of various types of cancer cells along with malaria, anthrax, and dengue. More recently, Chan et al. have reviewed microfluidic platforms employing DEP with respect to a particular cancer cell type [191]. Furthermore, this review also introduces the basics of DEP theory and CTCs formation. Menachery et al. have reviewed microfluidic label-free methods for stem cell isolation including those based on electrical cell characteristics [227]. Li and Anand have reviewed approaches of dielectrophoretic cell manipulation combined with a single cell analysis including both on-chip and off-chip realizations [228]. Nuchtavorn et al. have described a different separation approaches and devices employed for microchip electrophoretic manipulations with various biomedical analytes like drugs, nucleic and amino acids, peptides, proteins, antibodies, antigens, cells, and cell components [229].

##### Magnetic Cell Separation

The application of the external magnetic field is one more reliable way to induce the forces responsible for highly selective cell manipulations. This is achievable since biological materials have a very low magnetic susceptibility, and thus, the cells labeled with magnetic particles or having their natural magnetic response can be effectively isolated without the interference of the surrounding medium (see Figure 9b) [156]. Therefore, the essential prerequisite of the magnetic sorting is a high magnetic response of the objects to be isolated (or its absence in case of the negative sorting). If the cells of interest do not possess the high magnetic susceptibility itself (for instance, as the red blood cells), one should consider to label them with an appropriate magnetic tag.

Since superparamagnetic nanoparticles have a natural magnetic polarization, a non-uniform magnetic field (or a magnetic field gradient) is required to move them by magnetophoretic force analogous to the dielectrophoretic separation. Thus, all magnetic separation methods employ the structures creating local inhomogeneity of the magnetic field acting as trapping or deflecting sites for the cells labeled with magnetic particles. This was successfully realized by Miltenyi et al., who first proposed in 1990 a parallel cell separation technique for routine use in laboratory currently known as a conventional magnetic-activated cell separation or MACS [230]. They used a separation column filled with ferromagnetic steel wool placed between the poles of a permanent magnet. The steel wool fibers disrupt the uniform magnetic field between the magnet poles whereas the dense wool packing reduced the distance between the cells and the magnetic field source. These two factors induced a high local magnetophoretic force that effectively captured the labeled cells. The cells of interest were labeled with the fluorescent tag and superparamagnetic nanoparticles by streptavidin-biotin coupling.

Zborowski’s group has suggested an alternative way for continuous flow magnetic separation developing a quadrupole magnetic flow sorting approach (QMS) [226,227]. Generally, QMS exploits an annular flow channel with a spherical section placed in the radial gradient of the quadrupole magnetic field oriented from the core to the walls of the channel (Figure 13).

The main benefit of the quadrupole field is the fact that magnetophoretic force acting on the particle is finely defined by the particle position over the channel cross-section. The QMS separation concept is based on the cell magnetophoretic mobility that is the velocity of the cell per unit of the magnetic force. Passing through the channel, the cells labeled with magnetic nanoparticles are deflected to the channel walls, whereas less mobile unlabeled cells remain near the core rod forming two outflows separated by a flow splitter. The inlet flow of the initial cell mixture is also separated from the channel wall by the flow splitter to avoid the non-labeled cells pass near the magnet and come to the positive outflow with the magnetically isolated cells. To date, QMS has been successfully employed for positive selection of magnetically labeled CD34+ blood progenitor cells from blood circulation [232]; depletion of immunomagnetic labeled T-cells for clinical allogeneic bone marrow transplants [233]; separation of magnetically labeled porcine islets of Langerhans for further transplantation in diabetes treatment [234,235]; and isolation of intrinsically magnetic, deoxygenated RBCs from the whole blood [231].

However, modern tendencies in magnetic cell separation are mostly related to the development of microfluidic sorting systems. The employment of the microfluidics allows for the relatively simple generation of local high gradient magnetic fields in the desired area of the microchip and preparation of microfluidic channels of various shape and geometry by well-established microfabrication techniques. Additionally, microfluidic systems significantly reduce the required volume of the analyzed sample. Novel microfluidic platforms for magnetic cell separation are constantly developed, numerically simulated, and tested for continuous-flow separation of magnetic beads (for instance, [236,237,238,239,240]). Furthermore, these systems were shown to be effective for the isolation of magnetically labeled cells including rare circulating cells from blood samples. As a recent example, Shi et al. successfully isolated HCT-116 human colorectal cancer CTCs selectively labeled with magnetic nanoparticles from the whole blood samples by a wavy-herringbone structured microfluidic device with a capture efficiency of 81%–95% [241]. The specific wave structure of the microchannels creates turbulence in the liquid flow increasing the colliding probability with the channel walls and improving the effectiveness of the cells capturing. Later, Zhang et al. employed the microfluidic device with a twin-layered herringbone structure for immunomagnetic isolation of Hep3B CTCs from blood samples with a capture efficiency of about 90% at the clinical relevant tumor cell density in blood [242]. The specific structure of the microfluidic channel was also exploited by Jung et al. [243] They combined magnetic separation device with slanted ridge array creating advective rotational flows delivering magnetically labeled cells precisely to the area where high magnetic forces are formed. This enables us to separate the cells dispersed in whole blood at flow rates up to 0.6 mL/h with an efficiency of about 90%. Shen and Park described a microfluidic magnetic separation system that can separate subpopulations of macrophage Raw 264.7 cells depending on their absorption capacity [163]. Xu et al. developed a microfluidic setup for high purity isolation of CTCs from the whole blood [244]. They demonstrated effective immunomagnetic labeling of MCF-7 human breast cancer, SGC 7901 human gastric cancer, Hela human cervical cancer, and PC3 human prostate cancer cell lines with magnetic composite particles along with isolation from RBCs and leucocytes. Droz et al. demonstrated an effective approach for isolation of the cells expressing proteins of interest employing automated microfluidic setup [245]. This approach deals with some issues related to the realization of automated cell sorting like homogeneous dispersion of magnetic nanoparticles in the microchamber for effective cell labeling, various rates of the protein secretion by the cells, and effective isolation of monoclonal cell populations. Lee et al. successfully isolated CTCs from the blood samples of cancer patients by one-step negative enrichment of white blood cells labeled with magnetic nanoparticles [246]. They exploit a microfluidic multi-vortex mixing module in the microfluidic setup for effective white blood cell labeling. An analogous approach was used by Bhagwat et al. to pre-enrich rare circulating cells from the whole blood by in-line magnetic particle-based leukocyte depletion prior to flow cytometry analysis [247]. Huang et al. designed a microfluidic microwell device that allows for immunomagnetic single-cell trapping for further cellular analysis at a single-cell level [238]. Green et al. described a microfluidic magnetic sorting device that allows for the effective capturing of CTCs and dividing them into subpopulations according to levels of protein surface expression [248]. The analysis of subpopulations of CTCs that exhibit different biochemical and functional phenotypes may be useful to figure out if the CTCs acquire the metastatic potential by undergoing the epithelial-to-mesenchymal transition (EMT).

An important aspect of magnetic cell sorting is the isolation of the cells with an inherent magnetic moment. There are several reasons to exploit the label-free separation of intrinsically magnetic cells rather than utilize labeling approaches. One of them is purifying biological samples from an excess of RBCs for diagnostic or therapeutic applications. For instance, isolation of RBCs form stem cells is desirable at bone marrow transplantation [249]. Additionally, it is known that blood transfusion may be associated with increased side effects due to the damage of RBCs after long storage [250,251]. Another critical issue is the isolation of RBCs for further analysis. Therefore, it is highly desirable to avoid additional contamination of the cells with antibody-conjugated magnetic particles. Melville et al. and Owen performed the first experiments devoted to RBCs magnetic isolation from the whole blood confirming their paramagnetic properties and proving the concept of label-free RBCs separation from the whole blood by applying high gradient magnetic field [252,253]. Takayasu et al. described continuous flow magnetic separation of RBCs from the whole venous blood in a glass tube [254]. Later on, magnetic separation of RBCs was redesigned with respect to microfluidic techniques (for instance, Han and Frazier [255] and Qu et al. [256]). As a more recent example, Myklatun et al. developed a setup for microfluidic sorting of intrinsically magnetic cells under visual control [164]. The setup was shown to be effective for contamination-free isolation of intrinsically magnetic cells (magnetotactic bacteria) with the possibility of automated quantification of the separated cells. Zborowski and Chalmers with co-workers also have studied the separation of intrinsically magnetic cells. They figured out the difference in magnetophoretic mobility of oxygenated and reduced RBCs [257], developed a theoretical model of microfluidic RBCs sorting [258] that was used as a basis for QMS device for continuous RBCs sorting [231].

Furthermore, the development of continuous flow magnetic isolation of infected RBCs is of great importance with respect to malaria treatment. Malaria is an infectious disease caused by *Plasmodium* parasites affecting host RBCs. During the life cycle, the parasites metabolize most of the cellular hemoglobin to hemozoin exhibiting paramagnetic properties in contrast to oxygenated hemoglobin, which is diamagnetic [259]. Moreover, recent research reported on the superparamagnetic behavior of hemozoin nanocrystals [260]. This difference in the magnetic response of healthy and parasite affected RBCs can be employed for magnetic separation. Thus, a decent number of separation approaches were developed. Paul et al. first reported on the batch separation of malaria-infected erythrocytes from whole blood using a high-gradient magnetic separation [261]. Moore et al. studied the magnetophoretic mobility of infected RBCs and demonstrated that their magnetic properties change with respect to the development stage of the parasite [262]. Further, Nam et al. developed a microfluidic device for continuous-flow separation of malaria-infected RBCs in various stages of development [263]. They reported on up to 99% efficiency of the separation of the late-stage infected cells and the possibility to isolate early-stage developed parasite-infected cells. As for more recent results, a number of works are performed [264,265,266], devoted to numerical simulation of microfluidic devices to magnetically isolate malaria-infected RBCs. However, a possible way to make magnetic separation to meet the demands in clinical scale blood purifying system is a building up of a new efficient high-throughput mesoscale separation devices as reported by Martin et al. [267] They described a continuous flow high-gradient magnetic separation device for isolation of infected erythrocytes from the whole blood without a need for shear flow or blood dilution. In this design, the blood can be taken directly from the patients and afterward returned upon purification. This system is reported to be about 380 times more efficient comparing microfluidic systems. To complete this passage, we refer to a recent review of Kasetsirikul et al. covering malaria diagnostic methods along with techniques for malaria detection and infected RBCs separation including DEP and magnetophoretic approaches [268].

Despite the current popularity of microfluidic approaches for magnetic cell separation, these methods have a considerable drawback that is a small volume of processing blood samples and thus the separation of rare circulating objects (e.g., CTCs). Although microfluidic isolation can almost completely remove the target objects (with efficiency about 90%–95%), a typical sample volume for microfluidic investigation is about 5–10 mL, and this can only contain a tiny amount of rare blood biomarkers that is insufficient for adequate analysis and reliable diagnosing [269]. Therefore, alternative approaches for fast processing of large blood volumes including in vivo analysis, are being developed. Vermesh et al. described an immunomagnetic cell enrichment method for the isolation of CTCs in vivo [270]. This implies the immunomagnetic CTCs labeling with antibody-conjugated magnetic beads injected directly to the bloodstream and capturing of the labeled cells with a flexible magnetic wire that was inserted and removed through a standard intravenous catheter. Although the capturing efficiency of the system in a model experiment in vivo was shown to be only 8%, it was able to capture from 2500 to 10,000 cells per minute and these are 10 to 80 times more efficient comparing to microfluidic platforms.

Finally, we refer to some recent reviews covering various issues related to magnetic cell sorting. Plouffe et al. have reviewed the application of magnetic particles for cell isolation [157]. This includes an overview of biomedical applications utilizing magnetic separation and various cell separation approaches; a theoretical background of magnetic phenomena and nanoparticle properties that can be employed for the cell separation, including synthesis methods of appropriate magnetic particles; and examples of the setups for manipulation with the cells conjugated with magnetic nanoparticles. Huang et al. have reviewed progress in magnetic manipulation with particles and cells in microfluidic chips including their separation, concentration, capture, arrangement, and assembly [271]. The review also includes the theoretical background of particle and cell manipulation in a microfluidic environment. Munaz et al. have reviewed advances in magnetophoresis with respect to microfluidic systems, including fundamental and theory of magnetophoresis and its applications for mixing, separation, and trapping of particles and cells [272]. Pezzi et al. have evaluated five different types of commercially available magnetic beads with respect to various aspects of cell labeling and isolation [273]. They have tested antibody binding, and surface density, target cell capture efficiency and purity, the release of conjugated antibodies and captured cells, an impact of magnetic beads on further cell imaging and analysis, and the possibility of integrating magnetic beads cell isolation with standard nucleic extraction methods. This work may be used as a comprehensive guide to choosing the most suitable magnetic beads for cell manipulations depending on the requirements and possibilities of a particular investigation.

##### Acoustic Separation of Cells

Acoustophoresis is an application of the acoustic field for cell sorting (see Figure 9c). Acoustic waves in the ultrasound range are applied in a number of microfluidic devices [165] allowing for separation of cells according to their size, density, and compressibility (Figure 14) [274]. Acoustophoresis operates with an acoustic power and an ultrasound frequency similar to those of ultrasound imaging, and, therefore, does not influence significantly the cell viability [275,276] that opens prospects to use this technique for cell sorting.

Ultrasound waves used for acoustophoresis can be divided into standing and traveling waves [277]. In turn, standing waves can be classified on bulk acoustic waves (BAW) [278] and surface acoustic waves (SAW) [279]. The raveling acoustic waves occur when ultrasound propagates in the medium with more or less uniform acoustic impedance. The pulses of traveling surface acoustic waves may be employed for isolation of single cells from the main sample stream and divert them in another outlet of the microfluidic chip. This was shown to effective for the isolation of fibroblasts, keratinocytes, and melanoma cell lines [280,281].

The standing acoustic waves occur under resonance conditions. To perform acoustophoretic sorting with standing BAW, ultrasound is typically applied transversely to the cell flow direction. The standing BAWs will occur in a microfluidic channel if the ultrasound wavelength matches the channel width [165]. This kind of wave is used to redirect cells from the main streamline toward the pressure node to the special outlet. Currently, microfluidic chip technology combined with bulk acoustic waves is used for separation of different mammalian cell subpopulations, such as red blood cells, white blood cells, platelets [282,283], and cancer cells like neuroblastoma [283]; breast cancer [284,285], prostate cancer [278], pancreas cancer [286]. Moreover, this approach allows for separating of other types of blood emboli like a lipid embolus [287] and bacterial cell [288]. In turn, standing SAWs are formed in the microfluidic channel when ultrasound propagates through the longitudinal direction [289]. Standing SAWs may be applied for isolation of either normal cell lines like white blood cells [290], platelets [291], and muscle cells [292] or cancer cells like breast cancer [279,292], melanoma [280,281], and alveolar basal epithelial adenocarcinoma [292]. Furthermore, the tilted-angle standing surface acoustic waves are used for white blood cell sample washing from cell debris formed during red blood cell lysis procedure [293].

At present day, there are microfluidic devices employing both standing and traveling surface acoustic waves for the selection of cancer cells from the blood. This multi-stage sorter used standing waves for cell focusing in the central part of the channel and traveling waves for displacing of target cells to the desired outlet [294].

#### 4.3.2. Passive Cell Separation Methods

The sorting approaches described above include active particle or cell separation techniques. There are also passive methods developed for the separation, isolation, or enrichment of particles and cell populations based on typical differences in their size, shape, or morphological parameters. Inertial forces, hydrodynamic propagation, and deterministic lateral displacement are applied to sort different objects [295,296].

##### Inertial Focusing in Microfluidic Channels

One of the commonly used approaches for label-free passive cell separation is inertial focusing in microfluidic channels. The theoretical background and detailed description of the separation process were summarized by Di Carlo [297]. Briefly, the inertial focusing of the cells or particles can be achieved when the liquid flow in the microchannel is still laminar, but inertia becomes valuable. This flow mode depends on the geometry of the channel and liquid velocity and viscosity and usually observed from very low Reynolds number Re (Stokes flow, inertia is negligible) to ~2000 (lower range limit of turbulence flow). The cells moving in the flow are affected by drag and lift forces and therefore aligned in a defined equilibrium position along the main flow direction (Figure 15a). The equilibrium position is defined by the ratio of the drag and lift forces and varies with the third power exponent of the particle/cell diameter [298]. In this way, kinetic and equilibrium separation can be realized. The kinetic separation is based on the different equilibration times for the cells of various size whereas the equilibrium separation implies different equilibrium positions for different cells.

The curved microchannels may be employed to produce inertial forces. Considering the laminar inertial flow, the liquid flowing through the curved channel results in the re-circulating secondary flows that are formed due to a mismatch of the liquid velocity in the channel center and near-wall regions (Figure 15b). These re-circulating flows create a pressure gradient in the radial direction of the channel and form two symmetric vortices inducing secondary flow. The dimensionless number describing the secondary flow was first established by Dean and, therefore, called the Dean number, while the secondary flow itself is called the Dean flow. Thus, the cells in curved channels are affected both by inertial migration and Dean flow. This results in the movement of equilibrium line from the channel center that allows improving cell collection; to reduce the channel length required for focusing and separation; to enable an additional equilibrium separation modality for the cells of various sizes (Figure 15c,d).

As recent practical examples, Nathamgari et al. used curved microchannel to isolate single stem cells from cell clusters after dissociation from the initial tissue [299]. Son et al. employed inertial separation in the curved microchannel to purify sperm cells from RBCs and leukocytes [300,301]. Schaap et al. first demonstrated the microfluidic separation of three specimens of algal cells of various sizes and shapes, including non-spherical ones [302]. Lee et al. used inertial separation in the curved channel for mesenchymal stem cell enrichment from a tissue-digested mouse bone marrow cell mixture [303]. The further development of inertial focusing is related to the preparation of more sophisticated microchannels and the integration of curved channels with other separation techniques. For instance, Shen et al. demonstrated spiral microchannel with series of micro-obstacles inside allowing for the linear acceleration of the secondary flow (Figure 16a) [304]. The acceleration can be tuned employing various geometry of the obstacles and degree of confinement. This results in increased efficiency and throughput of separation. Nivedita et al. combined the spiral channel with a lateral cavity acoustic transducer [305]. The microchannel provided a passive separation of non-target cells while the acoustic transducer is used for further active enrichment of the purified cell sample. Ramachandraiah et al. developed the method for effective isolation of leucocyte subpopulations from the unprocessed whole blood samples based on the combination of inertial microfluidics and cell lysis. The reported device allows for complete removal of the RBCs by osmotic lysis along with the separation of the nucleated cells depending on their size in the curved microfluidic channel. Zhou et al. designed an asymmetric reverse wavy microchannel and successfully employed it for the isolation of cancer cells from a blood sample (Figure 16b) [306]. This channel geometry induces periodically reversible Dean flow leading to effective size-depended particle and cell separation. Syverud et al. demonstrated the microfluidic device with «Labyrinth» patterned curved channels [307]. The device was shown to be effective for the isolation of myogenic cells from the enzymatically processed muscle tissue cell mixture and for the purification of them from cell aggregates and debris. Wang et al. developed a microfluidic separation device consisting of a wavy microchannel structure followed with a sheet flow-focusing section [308]. The introduction of the sheet flow-focusing area allows for more precise control over particle/cell position and even control the spacing between the separated objects. Furthermore, the concept of multisectioned devices can be extended for preparation of automated integrated microfluidic devices containing several focusing and separation areas. Zhang et al. described the example of the device containing flow regulatory and two focusing areas in consequently connected microfluidic chips (Figure 16c) [309]. The focusing was performed in spirally shaped microchannels. The device was also equipped with a set of operational and control units to carry out fully automated cell manipulations and was shown to be capable of efficient separation of cancer cells from human blood.

However, the preparation of curved and wave-shaped microchannels might be challenging, and, for some applications, the straight microchannels are preferable due to the ease of preparation and parallelization. Thus, one of the possible approaches to improve inertial separation with straight channels is the employment of the channels with an asymmetrical cross-section. For instance, Moloudi et al. investigated the possibility of particle separation in a trapezoidal straight microchannel [310]. They figured out that size-dependent particle separation could be effectively controlled by the slope of the channel wall and the speed of the carrying fluid. Another issue that may be addressed with straight microchannels is the isolation of small pathogens and blood components like fungi or exosomes. However, the conventional inertial focusing of such small bioparticles requires the significant length of the microchannel and a significant amount of time for processing. For this purpose, the concept of oscillatory inertial focusing in the practically infinite channel was proposed. In particular, the direction of liquid flow in a symmetrical microchannel is switched to the opposite one with a high frequency. Due to the symmetry of the velocity field, the particles preserve their motion direction, which allows for continuous separation like in infinite channels. For instance, Mutlu et al. demonstrated the successful isolation of submicron particles and *Staphylococcus aureus* bacteria exploiting this approach [311].

##### Deterministic Lateral Displacement

One more separation approach that can be employed to separate particles or cells in continuous flow mode is deterministic lateral displacement (DLD). DLD is a method that utilizes an array of posts placed in the microchannel in a defined manner that is every following raw is shifted latterly toward the previous one at a set distance [312]. This leads to the formation of separate laminar streamlines that follow through the structure in a certain path. If the particle size (R_eff_) is more than a critical radius (R_c_), the particle moving around the post will be deflected to the neighbor streamline [313]. This action repeats in the vicinity of every post that results in the displacement of the particle from the initial trajectory with a displacement angle determined by the row shift (“bumping” mode). Otherwise, the particles smaller than R_C_ remain in the same streamline and keep following a “zigzag” but eventually straight path (see Figure 17a). The critical particle radius is defined by the post diameter, the distance between the posts, and the raw shift [313].

First, DLD was introduced as a separation technique by Huang et al. [314] They demonstrated effective isolation of two types of fluorescent beads passing through the post array depending on single bead critical size based on the fixed gap between the posts. Additionally, they have shown the possibility to resolve the spectrum of particles from 0.8 to 1 μm by passing through the separation device containing several separate sections with various gaps. Later, Inglis et al. provided an extended theory of DLD separation supported by experimental measurements for a range of particles from 2.3 to 22 μm [313]. Further, the same group employed DLD for blood cell separation [315]. The authors described two types of DLD separation devices, allowing for separation of blood cells depending on their size or preparation of blood plasma i.e., to remove the cells from the whole blood completely. DLD was shown to be effective for isolation of lymphocyte subpopulations [316], nucleated RBCs from the peripheral blood of pregnant women [317], *T. cyclops* parasites from RBCs in whole blood [318], and cardiomyocytes form heterogeneous cell mixture [319]. Further development of DLD was related to the improvement of throughput of DLD devices to extend the volume of the processed sample per time to unite and enable the rapid enrichment of rare circulating objects like CTCs. For instance, Liu et al. proposed the DLD microfluidic device to isolate CTCs from the peripheral blood with a throughput of up to 2 mL/min [320]. They figured out that the CTCs isolation efficiency depends on the shape of the post. The triangle posts induce less shear stress and, therefore less effect on the cell hydrodynamic radius compared with the round ones. Thus, the cells keep moving through the streamlines as defined by the DLD pattern that improves the separation efficiency. Moreover, the same group demonstrated that the purity of the isolated cells could be further improved if DLD is combined with affinity-based separation techniques [321]. Previously, Loutherback et al. demonstrated the DLD device composed of triangle posts that can be run with a throughput of 10 mL/min and isolate CTCs from the whole blood with an efficiency of 85% [322]. However, the increase of the flow rate may lead to undesired side effects in the DLD array. One of the most significant is the clot formation when processing the blood samples. Although Loutherback et al. reached the significant flow rate in their device, the volume of the processed blood was limited by 200 μL per DLD array due to the clogging. D’Silva et al. carried out the study of clot formation in DLD arrays under various conditions and pointed outed how the clots can affect the DLD performance [323]. The clots increase the fluid resistance, capture the cells passing through the post array, and alter the paths of the fluid streamlines. According to D’Silva et al., clogging in DLD array may be induced by two factors: platelet the activation and coagulation that can be suppressed by inhibiting the action of thrombin and chelating of calcium ions respectively. Additionally, the authors demonstrated that shear-induced platelet activation is an important factor of clogging in microfluidic systems. This issue can be addressed by optimization of DLD array configuration, in particular in the injecting area. For instance, Mehendale et al. proposed the design of DLD device with a radial arrangement of round posts (RAPID) of various size combined in 3 separation zones [324]. This design was shown to be effective for continuous separation of spectrum of particles with respect to their size with throughput of 3 mL/min. The clogging was avoided due to preferential isolation of the largest particles while the rest of the particles can follow multiple parallel paths through the device. Further, they adapted the device for processing of whole blood [325]. Without any pre-processing, the device was able to operate with the whole blood for 6 h and rich almost 60-fold enrichment of the platelets as model target cells.

Another issue is that DLD, as other microfluidic passive separation approaches, implies a laminar fluid flow with Re less than 1. This is the pre-requisite for the formation of the defined laminar streamlines and a predictable movement of the particles through the post array. This results in a relatively low fluid flow rate and restricts the separation performance. On the other hand, the increase of the fluid velocity at high flow rates leads to the domination of inertial forces that alter the streamlines, which resulted in the formation of the vortices in the wake of the posts. The vortices increase the hydrodynamic radius of the posts reducing the critical radius of the particles. Recently, Dincau et al. reported on the numerical simulation and experimental evaluation of the performance of the DLD device operating in the high-Re mode (10 < Re < 60) [326]. They figured out that the vortices appear around Re = 25 and are fully developed under Re = 50, reaching the half-size of the post. Thus, the same particle can move through the DLD array in a “zigzag” mode under Re = 0.1 and in a “bumping” mode under Re = 25, which leads to lateral displacement. This may be employed for the development of DLD devices with an adjustable gap between the posts with the separation threshold tuned by the fluid flow rate. Further studies revealed that the main mechanism responsible for switching the displacement mode is associated with the compressing of streamlines under high-Re [327]. This was shown by the separation of vortex formation and streamline compression using the posts of a symmetric airfoil shape that does not produce the vortices even under Re = 100.

Additionally, DLD may be employed for separation of particles and cells by their shape involving two types of phenomena: differences in length scale and differences in transport properties [328]. For instance, Beech et al. combined size-based sorting with the isolation of cells regarding their susceptibility to shear stress (Figure 17) [329]. Further, Kabacaoğlu and Biros theoretically and experimentally studied the deformability-based sorting of same-size RBCs via DLD [330]. They provided an extended theory of RBC separation based on their deformability in DLD array and figured out that the deformability of RBCs is defined by their membrane stiffness and interior fluid viscosity, which in turn depends on the dimensionless capillary number and viscosity constant. Zeming et al. demonstrated complete isolation of RBCs to a focused stream employing an array of I-shaped pillars inducing continuous rotational movement of the cells that results in a greater effective separating size [331]. Au et al. designed the two-stage DLD device for the separation of CTC clusters with respect to their size and shape asymmetry [332]. The first stage of the device isolates the large CTC clusters with a size of 30 µm or higher from the whole blood moving through the conventional DLD array with the round posts. The second stage processes the product coming from the outlet of the first stage that includes small CTC clusters and blood cells using the array of asymmetric posts and height restrictions. The resulted isolation product contained about 99% of the large clusters and about 65% of small clusters recovered from the whole blood.

Although the design of DLD devices still is under consideration for every particular application, they were shown to be effective for cell isolation for cancer diagnostics. The examples include but not are limited to the works on CTC isolation mentioned above. Additionally, Wunsch et al. demonstrated the approach for the preparation of a nano-scaled DLD array that is suitable for the separation of exosomes [333]. More recently, Song et al. designed the DLD device with the posts modified by EpCAM aptamer covered Au NPs [334]. The aptamer-induced affinity capturing significantly increases the size-based enrichment of targeted CTCs in the DLD array. The captured cells can be released for further analysis by simple chemical reactions.

To conclude this subsection, we refer to the review of McGrath et al. devoted to the DLD separation [312]. This review provides a deep insight into the theoretical background of DLD principles, as well as the number of examples of various DLD devices, and can be used as a guide to assemble the DLD device for the desired application.

##### Pinched Flow Fractionation

Pinched flow fractionation (PFF) is an approach for the separation of particles according to their sizes in laminar flow (Figure 18). It was firstly proposed by Yamada et al. [335] and included two elements. The first is hydrodynamic focusing of particles in a fluid stream similar to the flow cytometry, but in this case, particles or cells additionally aligned against the microfluidic channel wall. Through this process, a flowing stream with particles or cells is pinched into a narrow channel cross-section-pinched segment. Once the channel broadens, the particles are forced to flow in different trajectories due to their size determines how far from the wall, their center of mass is placed. The second element is the flow field fractionation. Different outlet channels conjoined to the main channel with particles pinched to the wall. Due to adjusted fluidic resistances, the particles of different sizes are sorted by radially spreading streamlines and follow into different outlet channels. Particle separation efficiency is defined by the ratio of flow rates, the width of the pinched segment, the total flow rate, and the geometry of the segment collecting the particles. The parameters listed above should be adjusted in order to find the optimal design.

Yamada M. et al. separated polystyrene microspheres with a diameter of 15 μm from 30 μm ones [335]. The authors suggested a linear relationship between the particle radius and position in the pinched segment, assuming that all channels were much larger in width than in height. The pinching is taking place in the pinched segment with width 2d that should be similar to the diameter 2R max of the largest particle to be sorted. Therefore, the particles with a size significantly smaller than the channel width cannot be effectively separated. In order to overcome this problem and increase the particle displacement after passing the pinched segment, the geometry of one or more of the outlet channels was made asymmetrical to modify flow resistance for better hydrodynamic control. This realization was proposed by Takagi et al. as asymmetric pinched flow fractionation increasing the separation resolution [336]. This modification is capable of separating 1 μm and 2.1 μm particles with an efficiency of 80%. Additionally, the broadening of the pinched segment allows adding downstream collecting channels in the PFF design [337].

Alternatively, the separation resolution was enhanced by a simple geometric modification of the primary PFF design like adding a snakelike part to channel [338]. This results in a more effective separation of particles due to an increase of the lateral distance between the particles after the pinched segment. The enhanced PFF was shown to be capable of separating 7 different sizes of particles with an efficiency of 70%. In addition, PFF has been expanded by the spatial reorientation of the microfluidic device for gravitationally enhanced separation of particles with different sizes and mass [339]. The theoretical and experimental studies on dynamic characteristics of laminar flows and hydrodynamic enhancement of particle separation were carried out. Polystyrene microbeads with different sizes were used as model particles to demonstrate rapid (<1 min) and high-purity (>99.9%) effective separation. As a result, a simple sorting system was developed to perform size profiling and following mass-dependent separation of particles using a combination of gravity and hydrodynamic flows and applied to separate polydisperse perfluorocarbon droplets emulsions.

The first step towards nano-scale separation was realized by Y. Sai et al. by separation of 0.5 μm from 0.86 μm polystyrene microspheres [340]. They suggested incorporating microvalves to change the flow resistance in the collecting outlets of the PFF device to enhance separation efficiency. Expectedly, the pressure in the outlet varies depending on the microvalve position that leads to changing the flow rate inside the channel and the path of the flow through the particles. Thus, by adjusting the valves, one can force particles of different sizes to move to different outlets. Additionally, it was established that the separation resolution depends on the microchannel wall roughness that is the particles with a diameter comparable with the wall roughness could not be separated by the PFF technique [341].

In concentrated samples, interactions between particles increased, which leads to decreasing the selectivity of the PFF system. Thus, the influence of concentration needs to be investigated. The quantitative influence of the pinching intensity in the balance between the requirements of selectivity and minimal dilution was discussed [342]. As a result, PFF was proved as an efficient technique to separate a semi-concentrated polydisperse suspension of microparticles into sub-populations with tiny overlapping.

Eventually, the PFF technology was extended to a passive hydrodynamic cell sorting for leukocyte enrichment [343]. In addition, PFF was used to separate cancer cells (LS174T colorectal adenocarcinoma) from white blood cells (WBC), with that the cells were separated at efficiencies above 90% for both cell types [344]. The size overlapping between cancer cells and WBC prevents high-efficiency separation, however, it can be improved using a difference in cell deformability.

Further PFF improvements can be developed by using other separation techniques in combination with PFF, for instance, sedimentation [345], centrifugal [346], optical [347], and dielectrophoretic forces [348]. Furthermore, inertial enhanced pinched flow fractionation by taking advantage of inertial forces was devised [349].

## 5. Automatic Processing of Cytometry Data

After the flow cytometry data is collected, a detailed analysis should be performed. At the beginning of flow cytometry development, the analysis was carried out manually [350] and, therefore, consumed a lot of time, while the quality of analysis depended on the operator qualification. Currently, a cytometry device usually provides a user with a toolset for automatic data processing. The data processing pipeline in flow cytometry usually consists of some signal preprocessing, generating a set of features for each object in the flow and then provide separation of object classes based on the combination of features (Figure 19). Signal preprocessing is highly specific for each device design and, thus, it is outside the scope of this review.

Conventional flow cytometry methods provide an intrinsic set of features describing single objects in the flow. It usually includes forward scatter and side scatter channels along with signals from PMTs detecting fluorescence in a number of wavelength bands. Imaging flow cytometry, however, requires conversion from raw images of single objects into a set of features that allow discriminating properly different object classes. Feature construction and selection of separation criteria for a long time was a complex procedure that requires high qualification and a lot of expertise from the device operator. Progress in flow cytometry increased both the number of cells in the experiment and the amount of information obtained from single cells, so the processing methods became more complex. On the other hand, modern progress in machine learning and deep learning methods start to make formerly complex procedures completely automatic and less erroneous in general.

In the conventional analysis of flow cytometry data, expert works with a set of features—the values that describe each object in the flow. The most basic is the intensity of fluorescence in different spectral channels along with front scatter and side scatter light intensities. The multidimensional space of features is usually orthogonalized by a compensation matrix that describes how to minimize the crosstalk between channels. Afterward, the feature space can be distorted by some transfer function (e.g., logarithm, biexponential, logistic, tangential) that provides cut-off for the outliers or compression of the rare high-intensity events into the space near the major data points. The essential part of the data processing is based on gating. The features that provide separation of object subpopulations are selected and plotted as 1D or 2D plots. Then the data points in a range (for 1D) or a region (for 2D) of values are selected manually on the plot by visually monitoring the points clusters and the data for objects subpopulation is extracted, so the next separation can be done on this subset.

To perform data analysis after receiving the signals, specific features should be determined. The combination of these features should characterize the cells of interest in the most accurate way. Until recently, classical computer vision (CV) and machine learning (ML) algorithms, such as thresholding (for feature extraction) or gradient busting and random forest (to solve classification problems) were mainly employed in the analysis of flow cytometry data [352]. However, there are a few examples of the original algorithms [353,354] to detect rarely seen labeled cells in the sample and to segment the nucleus, respectively. The introduction of these algorithms allowed us to automate the analysis of cytometry data with fairly good accuracy. To date, these approaches can be considered traditional and most widely used. Traditional ML and CV are currently implemented in the majority of cytometry data processing software, for example, IDEAS by AMNIS or FCS flow cytometry by De Novo software, which is often used with the MATLAB package for additional calculations. There are also free open-source software packages, like ImageJ [355,356] and CellProfiler [357,358,359]. The pipeline tends to be the following: single images from a cytometer are loaded to the software (for example, CellProfiler) where they are processed (for instance, brightness is adjusted and borders are highlighted) and placed into separate files. Then, the files are transferred to the input of some machine learning system built on the basis of some computational frameworks programmed in MATLAB, Python, or some other general-purpose computer language.

A lot of effort was made to define the automatic gating procedure for flow cytometry data [360,361,362,363,364] or clustering of similar phenotype on multidimensional data [365,366]. A number of machine learning approaches for automatic separation of subpopulations in the feature space was developed. The most well-known are SPADE, FlowSOM, and CITRUS methods along with the viSNE alone and combined with the previous three ones for prior dimensionality reduction.

Spanning-tree Progression Analysis of Density-normalized events (SPADE) [367] organizes the cells in a hierarchy of related phenotypes. It allows for high throughput processing of data volumes and draws a tree of different phenotypes/subpopulations of objects that can be easily understood, and extract subpopulations of cells with similar properties. The inputs for the algorithm are selected markers/features set, outlier density, target density and desired number of the clusters. The first and the last ones are the most important, the second and the third regulate how many cells are excluded and how many will survive the downsampling process. The last parameter governs when the algorithm will stop.

FlowSOM [368] is another approach to build Self-Organizing Maps on the dataset and organize it into a minimal spanning tree. This algorithm can use a complete dataset without downsampling and runs visualization procedure much faster than the previous method giving, in general, a similar result.

The third automatic approach for the detection of subpopulations from a flow cytometry dataset is based on the t-distributed Stochastic Neighbour Embedding (t-SNE) [369] method. The method builds a projection from the dataset with high dimensionality containing a large number of independent features into 2D space in a way that neighbor points are still nearby and separate clusters become well visible in the projection. In the flow cytometry, this method is usually called viSNE [370].

Information about automatic methods not using dating summarized in a number of comprehensive reviews, that include Aghaeepour et al. [371] and Mair et al. [372]

The next major step in the automation of cytometry data processing is based on transfer from manual feature construction to automatic. This process in computer science is heavily based on the transfer from classical machine learning algorithms, which separate the data by manually selected and thoroughly checked set of features, to learning approaches, which construct features by learning.

Most of the deep learning approaches were based initially on learning convolutions in the separate layers of neural networks (NNs) architecture. However, nowadays there are various and versatile sets of network architectures providing required functionality and the progress in inventing new ones is very fast. First successful approach to construct deep learning models for computer vision was mostly based on a deep network with a lot of layers in the structure forming a pairs from convolution layer that learns transfer methods to construct a features at each level of abstraction and fully connected layers keeping information as a set of weights of selected features [373]. The more modern neural network architectures can be based on only convolutions [374] or some kind of recurrence (RNN) providing the memory of previously processed data inside the network.

Any machine learning model still requires a dataset to be trained on. During the training, deep learning algorithms independently selects geometric features from the available data (a certain combination of curved lines, a combination of pixels of a certain brightness), followed by the derivation of abstract features from, for example, presence of a vacuole in the image or damage of cell nucleus [375]. As a result, the capabilities of flow cytometry are increased. Large arrays of images can be processed without distraction by a preliminary search and registration of significant features, making the process fully automated. Moreover, with the help of NNs, one can detect those significant features, which were not assumed by a researcher [376].

Deep learning approach allows us to solve some complex computer vision problems like semantic segmentation (separation of closely packed objects) or object detection with the accuracy of well-trained human experts and sometimes even better, however, it is achieved by the large size of required datasets. The usual balanced set of objects to train a deep neural network for computer vision problems should consist of tens of thousands of objects from each class. Sometimes it can be achieved by artificial augmentation of smaller datasets containing only thousands or even hundreds of objects, but it is mostly a required minimum to train a good working model. The initial marking of the dataset should be done by human experts and it is among the most tedious and complex tasks in a deep learning approach.

Additionally, a serious obstacle using deep learning methods in diagnostics may be their interpretability [377]. One or another step of the algorithm is difficult to explain in a large part of deep learning methods. In medical diagnostics, this may cause fatal consequences. The high system requirements for computing performance to run the deep learning algorithms is also can be considered as a disadvantage, however, it is now mostly eliminated by the computational power provided by the general-purpose graphics processing unit (GPGPU) or tensor unit acceleration with parallel processing.

Recently, there were a number of publications describing the complex system providing cell sorting by a combination of all the above-mentioned approaches. Nitta et al. demonstrated an integrated approach to the problem of finding the right cells and sorting them in a real-time combing traditional and deep machine learning [378]. Since neural networks usually have high accuracy, but relatively low speed, first for each event in the image a binary mask is created for the bright-field image, followed by extraction of the main characteristics from bright-field and fluorescent images. In this regard, algorithms of traditional computer vision were applied. For example, a median filter was applied to decrease image noise, whereas the Canny edge detection allows defining the contours of cells, particles or their groups. The extracted information about the geometry of the object allows building the binary mask. Further, the binary mask was applied to incoming images in order to receive information about their morphological characteristics (area and shape) and scattering and fluorescence intensity to highlight the region of interest. Afterward, the data was transferred to the input of a six- or eight-layer convolution neural network. This operation removes unnecessary information and makes it easier to extract intricate features. To put in other words, in the developed method the traditional ML and DL compensate the disadvantages of each other and combine high speed and accuracy. Testing the effectiveness of the method was done by sorting particles of different diameters in the sample (3-mm and 6-mm).

This paper also demonstrated a well-developed approach to building the entire computing architecture. Often this issue is neglected in research, giving preference directly to data analysis. However, an emerging trend towards the application of flow cytometers in diagnostics, engineering issues will have to be paid more attention. Properly built and optimized computing architecture can significantly increase effectiveness without the need for expensive equipment. In turn, this will make promising flow cytometer-based devices more affordable.

Li et al. addressed the problem of imaging cells without fluorescent agents to avoid their impact on cell behavior [351]. The authors developed an unusual approach to the problem of computational complexity. In particular, they discard most complicated operations, like image construction and extraction of features, and work directly with one-dimensional time series that display the shape of the detected wave. In order for the neural network could process with data, it was transformed into a two-dimensional array where one measurement corresponds to laser pulses at a point, the other corresponds to the number of points per pulse. Each signal was further divided into 100 smaller waveforms, with an overlap factor of 50% to create redundancy. This will further reduce noise and increase the sustainability of learning. After all the preparatory operations, the data were transferred to the input of a neural network with architecture based on VGG neural network architecture [374]. It consists of 16 convolutional layers, three layers for max-pooling, and three fully connected layers. As a result of the work, the network gives the probability of a cell belonging to one of three classes. The result was a score of 95.71% and accuracy 95.75% with high learning stability and recognition time within a few milliseconds.

Tanhaemami et al. predicted the lipid content in microalgae cells after nitrogen starvation by application of linear regression and a genetic algorithm (without the use of fluorescent labels) [379]. Eulenberg et al. [380] and Doan et al. [381] studied the cell cycle of Jurkat cells in diabetic retinopathy and morphological changes in red blood cells. Both papers mostly modify the existing model of machine learning and interpret conclusions in terms of biology, without reference to a specific device. The works employ training by standard images received from the cytometer without adjustment of the element base or the type of signals on the sensors. These examples demonstrate the flexibility of neural networks, as they require fewer preliminary data manipulation to work than traditional ML.

Certainty modern machine learning methods will gradually supersede traditional methods in the processing and analysis of flow cytometry data in the long run. ML and DL proved to be powerful and well-suited tools in cytometry. Probably, quite a few tasks can be solved by purely engineering approaches by adapting existing algorithms. Nevertheless, there is still the problem of further increasing the rate and biological interpretation of the results.

## 6. Advantages and Limitations of Modern Flow Cytometry

Flow cytometry technology has undergone many changes since its invention in the 50s of the last century. The development of new hardware and software for flow cytometry systems as well as fluorophores development expanded the number of science domains in which this technology could be applied. At the dawn of flow cytometry technology, it was used just to count and estimate the size of particles [382]. Currently, fluorescent flow cytometry systems are applied to solve a broad spectrum of issues in the diagnostics [383,384,385], biotechnology [386], material science [387,388], and other scientific fields.

The current state-of-the-art in fluorescent flow cytometry allows to estimate up to 20 independent parameters for each cell and possesses a high throughput possibility—thousands of cells per second [389]. However, the simultaneous usage of the number of fluorophores to label different cell structures can lead to the overlapping of their emission spectra and requires compensation procedure performance [390]. This step is crucial in the case of studying rare objects, such as CTCs, because their small amount does not allow an operator to create a compensation file. There are two approaches to solve this problem. The first one is to apply antibody capture beads to simulate positive, dim, and negative staining for the creation of a compensation file [391]. The second one is to use advanced software for automatic correction of the spectral overlapping [389].

Additionally, conventional fluorescent flow cytometry systems do not provide bright field and fluorescent images of events that constrain the ability of real-time verification. Imaging flow cytometry systems successfully solved this problem by combining the statistical power of conventional flow cytometry with the advantages of fluorescence microscopy [392]. The combination of both these technologies permits to acquire up to 12 fluorescent and bright-field images of each object without significant loss in throughput ability. However, this technology imposes some limitations on the object size, which is directly determined by the detection limit, dynamic range, and intensity resolution of the system [393]. Conventional flow cytometry also has the size threshold for object detection >500 nm [394,395]. One more disadvantage of flow cytometry is the disability to detect events in the whole blood due to its cell concentration [396] far more than the detection threshold (no more than 10^8^ cell/mL for Amnis, 3 × 10^7^ cell/mL for Bio-Rad, 7.5 × 10^6^ cell/mL for Beckman). However, a more detailed analysis of certain cell populations selected during flow cytometry measurement, for instance, genetic analysis requires some post factum procedure. With this regard, the demand for separation of the sample or its return for further analysis arises. Some devices such as Sony SH800, Bio-Rad S3e cell sorters, or Beckman MoFlo XDP High-Speed Cell Sorter have this ability, but this is the exception rather than the rule.

## 7. Conclusions and Outlook

The progress in the design of modern flow cytometry devices is strongly related to advances in adjacent areas of research and development. Considering the hardware part, great progress was made in the design of advanced optical systems like structured light illumination schemes, which allow for capturing better quality images on higher speeds of flow with less laser-induced damage of fluorophores. Additionally, the novel optical systems require fewer efforts in precise focusing of detected objects in flow and lens objective on the common spot. Great progress was also made in nonconventional detection methods including flow cytometry approaches for detection in vivo based on photoacoustic and photothermal phenomena. However, these substantial areas are out of the scope of this review.

Further, a semiconductor industry constantly provides better detectors that require less light to detect a signal. The modern techniques of image processing are able to take composite digital images by merging processed low-quality signals with properly removed noise from many sensors simultaneously. Semiconductor laser diodes and detectors for the less common part of the optical spectrum starts to be more available in all means and easier to integrate into new device designs.

Enormous progress in flow cytometry data processing software is achieved with the development of modern machine learning and deep learning techniques for all stages of data processing. The denoising and sensor fusion, solving computer vision and object detection tasks, grouping objects by the similarity of features that are automatically selected and generated are used everywhere now. All device and processing software vendors are starting now to provide this kind of solution to end-users.

With all this combination of novel techniques, the flow cytometry area of research is developing rapidly providing better devices that have better performance and much easier to operate. It pushes flow cytometry towards widely adopted endpoint-of-care devices from the research lab.

## Figures and Tables

**Figure 1 ijms-21-02323-f001:**
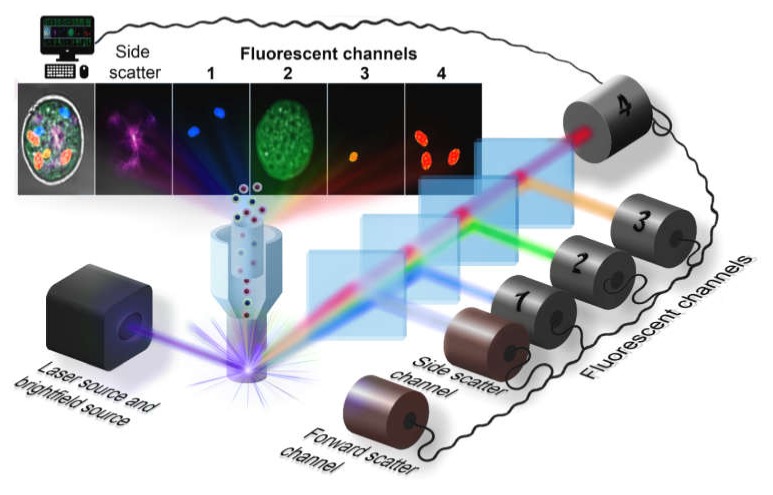
The optical system of an imaging flow cytometer.

**Figure 2 ijms-21-02323-f002:**
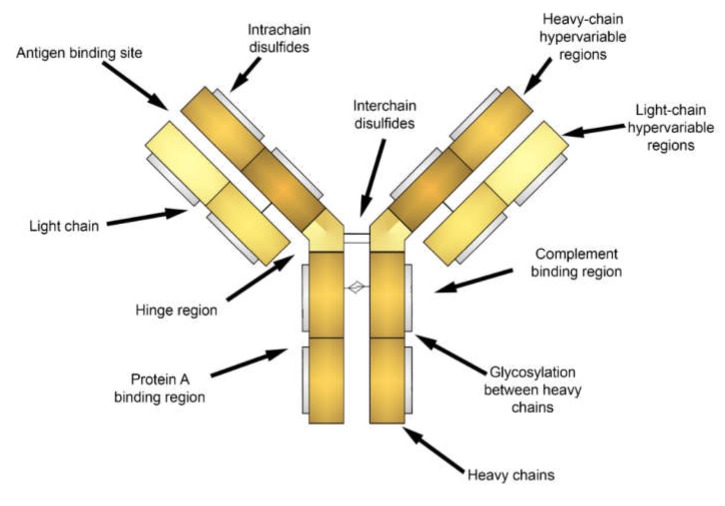
Sketch of an antibody structure.

**Figure 3 ijms-21-02323-f003:**
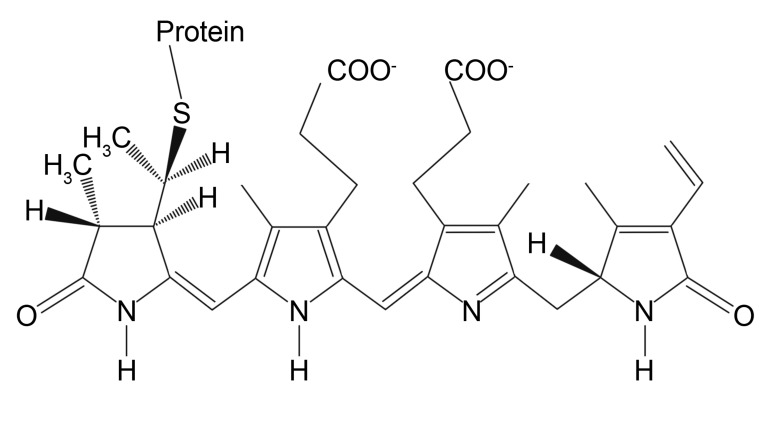
Chemical structure of phycoerythrin bilin chromophore.

**Figure 4 ijms-21-02323-f004:**
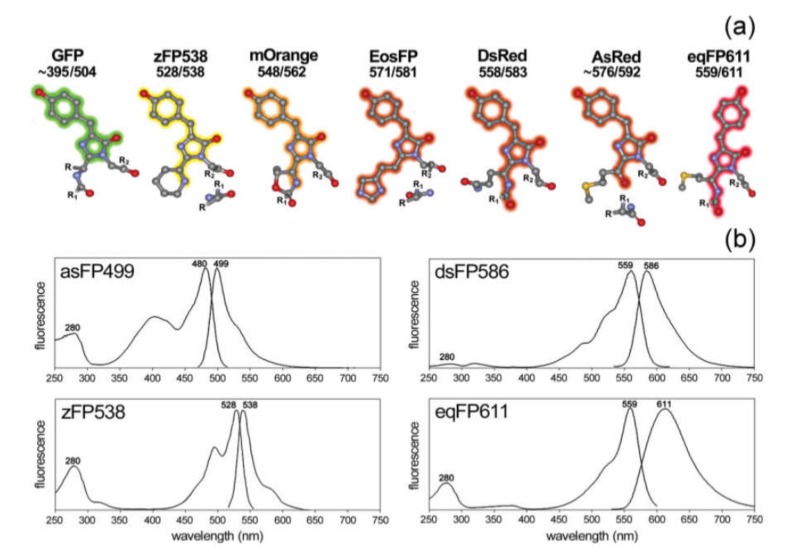
(**a**) Structural schemes of different chromophore groups (atom color coding: grey—carbon, red—oxygen, blue—nitrogen, yellow—sulfur; R/R_1_/R_2_ symbolize protein rests); (**b**) excitation and emission spectra of different fluorophores. Reprinted with permission from [141]. Copyright 2009, John Wiley and Sons.

**Figure 5 ijms-21-02323-f005:**
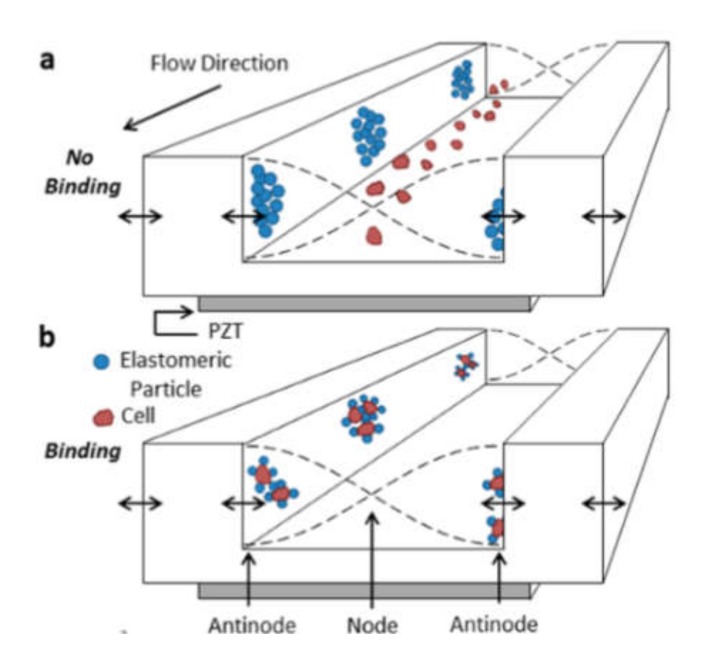
Distribution of: (**a**) elastomeric particles and cells and (**b**) cells binded with elastomeric particles in nodes and antinodes of the acoustic standing wave. Reprinted with permission from [166]. Copyright 2014, American Chemical Society.

**Figure 6 ijms-21-02323-f006:**
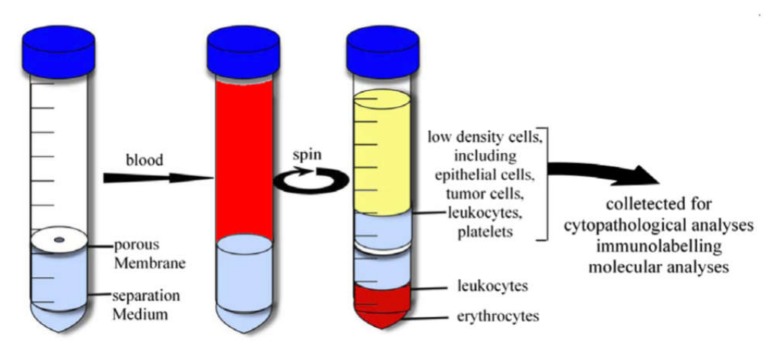
Cell separation by gradient centrifugation method with OncoQuick separation kit. Reprinted with permission from [180]. Copyright 2007, Elsevier.

**Figure 7 ijms-21-02323-f007:**
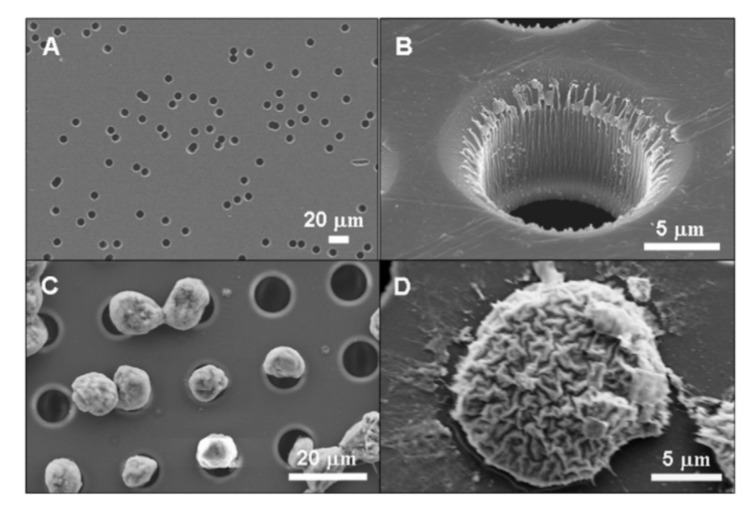
SEM images of (**a**) commercial membrane filter; (**b**) microfabricated parylene membrane filter; (**c**) parylenemembrane filter with cells captured without SEM fixation treatment, and (**d**) parylene membrane filter with cells captured after SEM fixation procedure. Reprinted with permission from [184]. Copyright 2007, Elsevier.

**Figure 8 ijms-21-02323-f008:**
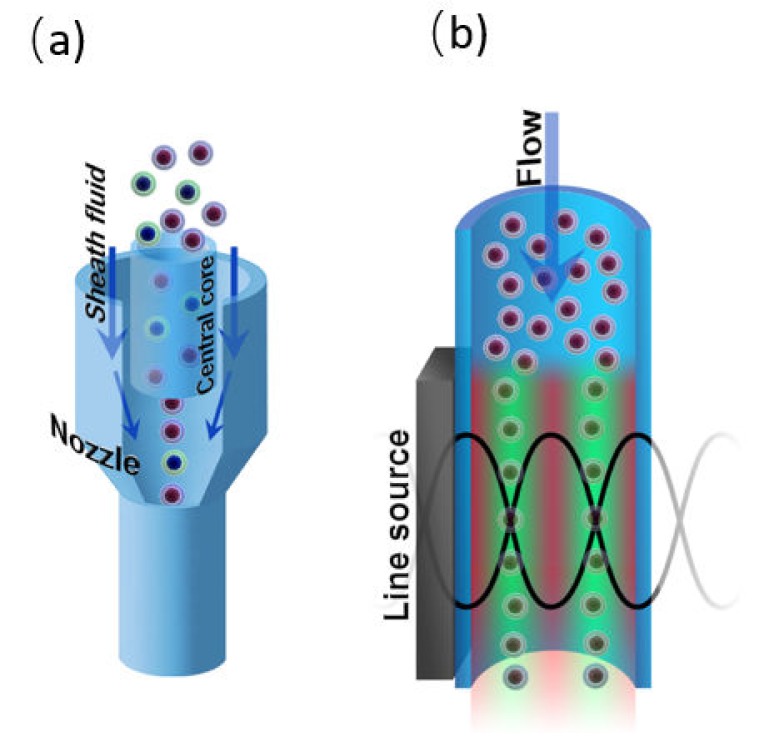
(**a**) Hydrodynamic and (**b**) acoustic focusing in a microfluidic channel.

**Figure 9 ijms-21-02323-f009:**
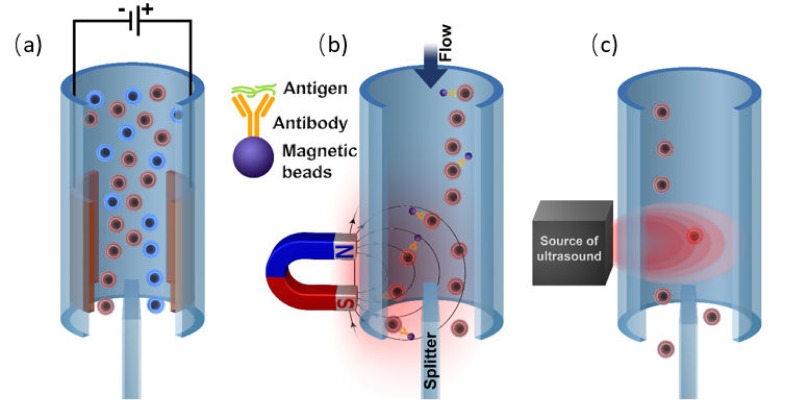
Various types of separation microfluidic mechanisms: (**a**) electrokinetic; (**b**) magnetic; and (**c**) acoustic separation.

**Figure 10 ijms-21-02323-f010:**
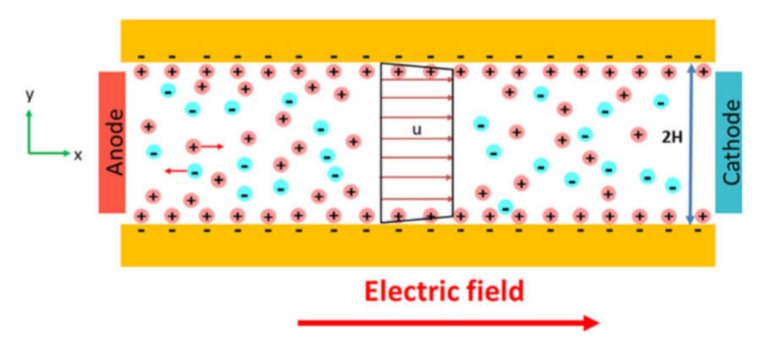
Schematic of conventional electroosmosis in a microchannel. Reprinted with permission from [190]. Copyright 2017, John Wiley and Sons.

**Figure 11 ijms-21-02323-f011:**
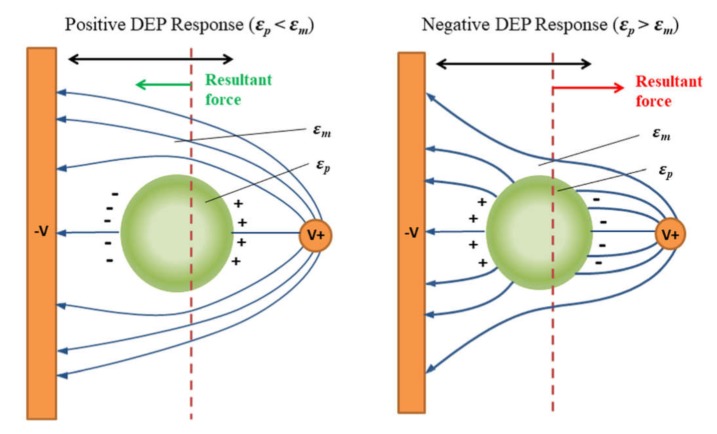
Scheme of the particle polarization process and dielectrophoretic response: left—positive DEP, right—negative DEP. Reprinted with permission from [191]. Copyright 2018, AIP Publishing.

**Figure 12 ijms-21-02323-f012:**
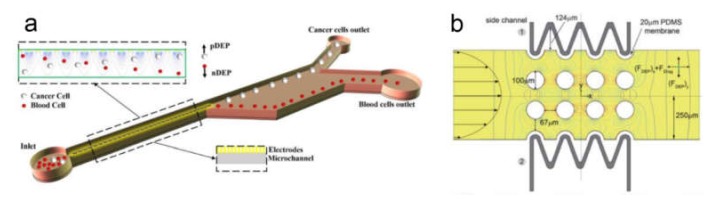
(**a**) Example of dielectrophoretic cell separation in the microfluidic device. Reprinted with permission from [199]. Copyright 2017, John Wiley and Sons. (**b**) Schematic of contactless dielectrophoretic (cDEP) device with the electromagnetic forces acting in the microfluidic channel. Reprinted with permission from [201]. Copyright 2010, the Royal Society of Chemistry.

**Figure 13 ijms-21-02323-f013:**
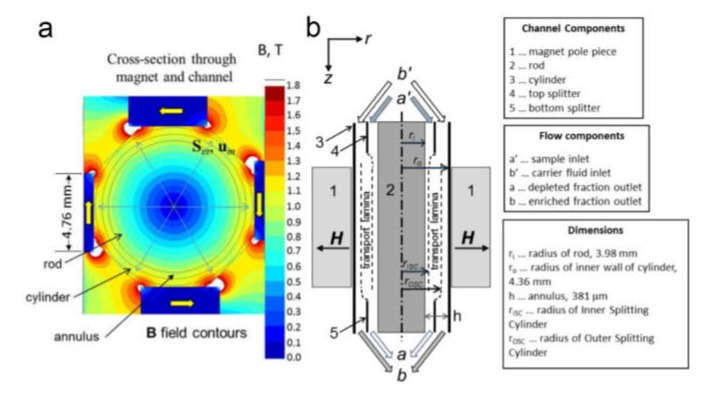
(**a**) Cross-sectional field map of the magnetic field of a quadrupole magnetic flow sorting (QMS) magent; (**b**) axial section diagram of the separation column contained within the quadrupole magnet assembly. Reprinted with permission from [231]. Copyright 2018, John Wiley and Sons.

**Figure 14 ijms-21-02323-f014:**
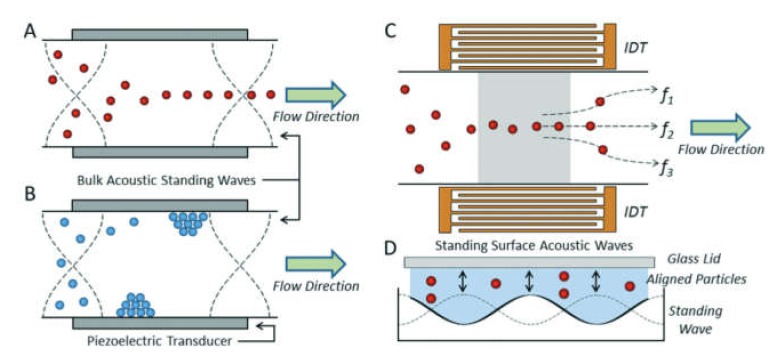
Acoustofluidic manipulations with particles and cells with a positive (**A**) and negative (**B**) acoustic response in a bulk acoustic standing wave. (**C**) The schematic of an surface standing acoustic waves (SSAW) device with interdigital transducers (IDTs) focusing the cells along well-defined streamlines. (**D**) The cross-section of an SSAW device with four pressure nodes. Reprinted with permission from [165]. Copyright 2015, Royal Society of Chemistry.

**Figure 15 ijms-21-02323-f015:**
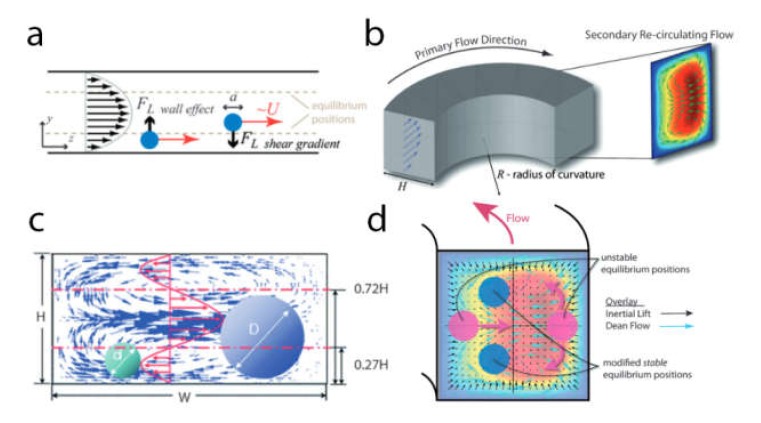
(**a**) Formation of the hydrodynamic equilibrium position of a cell moving in the liquid flow and affected by the drag and lift forces; (**b**) formation of re-circulating secondary flow (Dean flow) in the curved microchannel; (**c**) separation of the cells depending on their size employing secondary flow in curved microchannel; (**d**) the shift of the equilibrium position due to the superposition of inertial lift and Dean flow in a curved microchannel. Adapted with permission from [297] Copyright 2001, the Royal Society of Chemistry.

**Figure 16 ijms-21-02323-f016:**
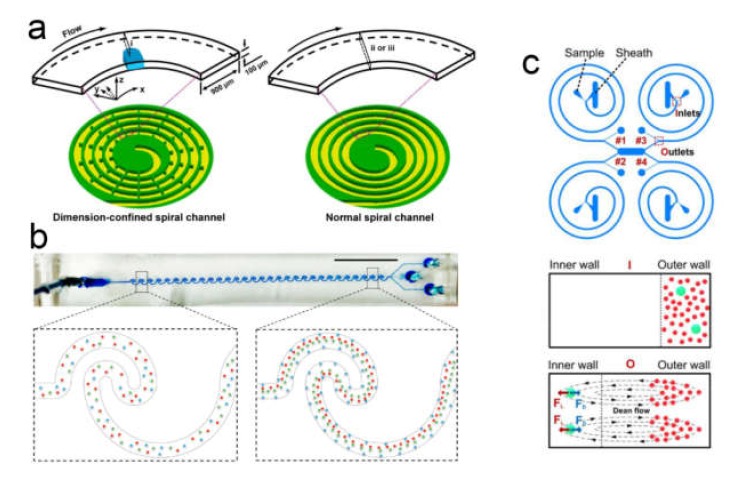
(**a**) Principal scheme of the device employing Dean-like secondary flow acceleration induced by micro-bars in the spiral microchannel. Reprinted with permission from [304]. Copyright 2017, The Royal Society of Chemistry. (**b**) Asymmetric reverse wavy microchannel inducing periodically reversible Dean flow. Reprinted with permission from [306]. Copyright 2018, Springer Nature. (**c**) Design of the separation device employing Dean flow with multiple separation areas. Reprinted with permission from [309]. Copyright 2001, The Royal Society of Chemistry.

**Figure 17 ijms-21-02323-f017:**
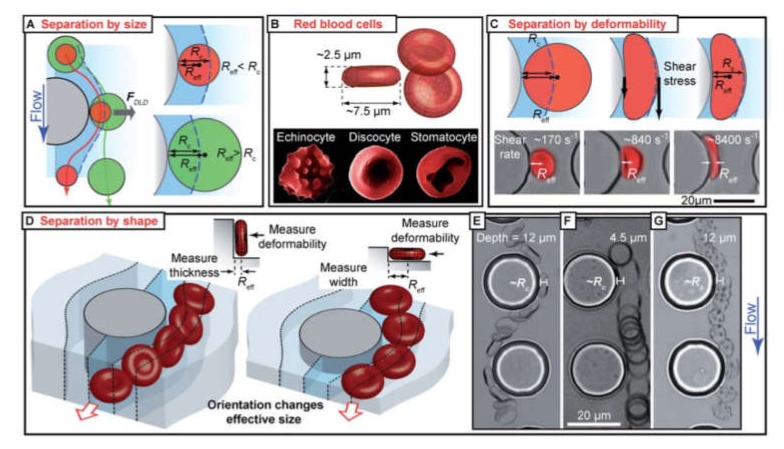
Mechanisms of separation by DLD with respect to cell size, shape, and deformability as in the case of RBCs. (**A**) The principle of DLD separation: particles with R_eff_ < R_c_ follow the flow direction and those with R_eff_ > R_c_ are displaced at an angle to the flow direction. For hard spheres, R_eff_ is equal to the radius. (**B**) Red blood cells are normally disc–shaped but they can adopt other shapes when exposed to different chemicals. (**C**) Shear forces deform particles changing R_eff_, and measuring the change in R_eff_ as a function of applied shear rate is equivalent to measuring the deformability of the particle. (**D**) R_eff_ depends on the orientation of the particle. Controlling orientation and measuring R_eff_ gives information about shape. It is also possible to measure deformability in different directions. (**E**) In a deep device RBCs rotate such that R_eff_ (< R_c_) is equal to half the thickness. (**F**) Confinement in a shallow device means that the cell radius defines R_eff_ (> R_c_). (**G**) An echinocyte with R_eff_ > R_c_. Reprinted with permission from [329]. Copyright 2012, the Royal Society of Chemistry.

**Figure 18 ijms-21-02323-f018:**
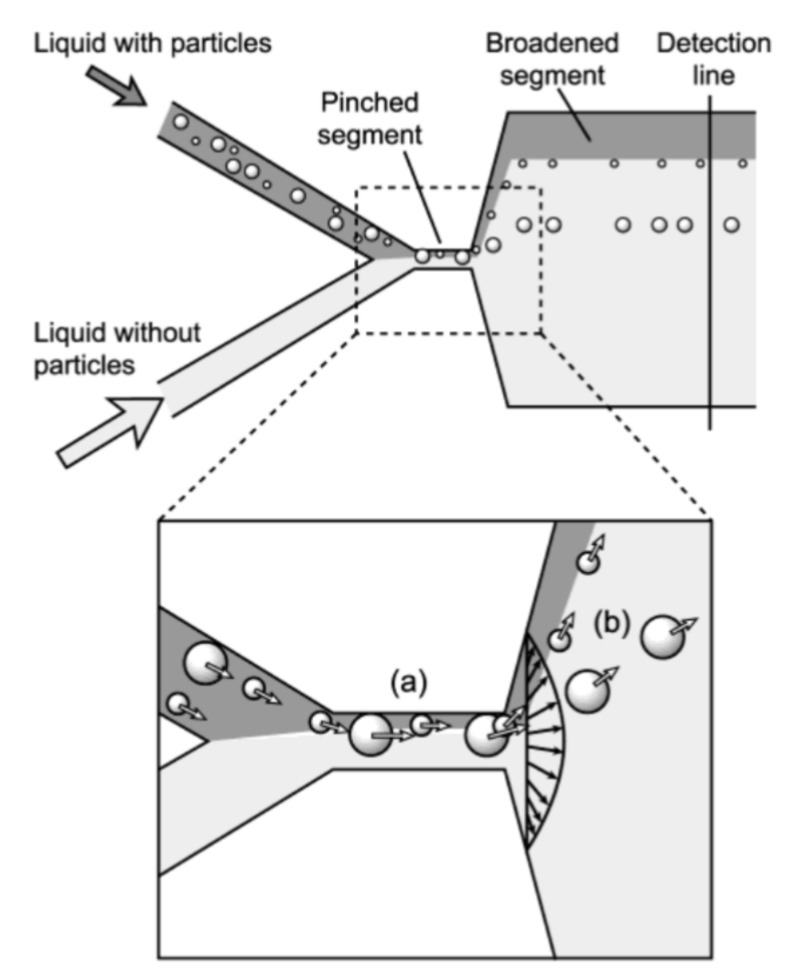
Principle of pinched flow fractionation. (**a**) In the pinched segment, particles are aligned to one sidewall regardless of their sizes by controlling the flow rates from two inlets; (**b**) particles are separated according to their sizes by the spreading flow profile at the boundary of the pinched and the broadened segments. The liquid containing particles is dark-colored. Reproduced with permission from [335]. Copyright 2004, American Chemical Society.

**Figure 19 ijms-21-02323-f019:**
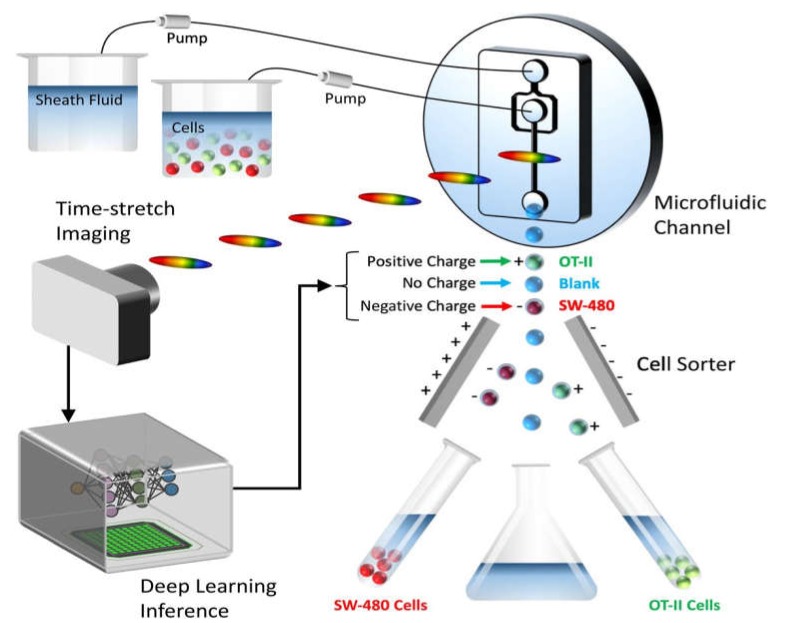
The deep neural network recognizes cell types by their corresponding waveforms. Different types of cells are categorized and charged with different polarity charges so that they can be separated into different collection tubes. Reprinted with permission from [351]. Copyright 2019, Nature Springer.

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
