# Peer review of "Detection of Rare Objects by Flow Cytometry: Imaging, Cell Sorting, and Deep Learning Approaches"

_ijms, 2020, doi:10.3390/ijms21072323_

Round 1
Reviewer 1 Report
This review is indeed very comprehensive with more than 350 referenced papers. It is generally well written with a few grammatical errors here and there, that do however impact the understanding of the document.
My biggest issue with this paper is that in its majority it talks about sorting of rare objects from a blood sample. There are a lot of pages on dielectrophoretic, magnetic, acoustic methods, as well as inertial focusing, deterministic lateral displacement and pinched flow fractionation, all of which are of course well established methods for sorting and isolating rare objects but not really related to the rest of the manuscript about flow cytometry (in the conventional meaning of the word).
The first many pages of the article talk about the lasers used, alignment, cameras etc. This is followed by a very thorough description of cell labeling. The first part is mostly focused on fluorescent particles/dyes etc while in the end some mention is given to magnetic particles or acoustic particles. Until section 2.2 everything that is written sort of suggests a standard flow cytometer. After 2.2 the focus seems to change completely to particle isolation and sorting techniques that have as such nothing to do with the previous part.
I would therefore suggest to the authors to create a much clearer red thread through the manuscript, e.g. by using some of the introduction to clarify what they are going to talk about and why.
Moreover, there are only 3 figures in the entire article, most of which provide very little information on the subject. Particularly figure 3 in no way conveys any sort of information on dielectrophoretic, magnetic and acoustic methods, other than the fact that all three focus particles one way or the other. Figure 2 is in principle also not very useful. The authors review more or less 350 papers, several of which can be pointed out for some clever solution. I would very much like to see a figure from some of these articles, either by asking for permission to use the figures from the articles or by redrawing the concepts in a new figure. Otherwise the article becomes very cumbersome to read, with too much text.
In page 25, line 1165 the authors say that inertia is observed from Reynolds number 0 to 2000. I would probably argue that you cannot have a Reynolds number that is zero.
Page 31, line 1484: Is it really 3 mm and 6 mm particles?
I suggest the authors change their title to not include "flow cytometry", as I believe it gives the wrong impression. Also, I would try to write something about the requirements necessary for a system like this to be of diagnostic value, in terms of e.g. throughput, analysis time, sensitivity. For some of the described systems, this information is highlighter, but it would be nice to add a table comparing the different techniques. Should the authors insist on talking about flow cytometry, I would also urge them to include a table with some commercial systems and what they are using of different techniques, what they are used for etc.
I believe that with some of theses changes this paper will be a valuable addition to the literature.
Author Response
We have duly considered the comments made by the Reviewer and corrected the manuscript text according to Reviewer suggestions. The major changes are marked with yellow.
This review is indeed very comprehensive with more than 350 referenced papers. It is generally well written with a few grammatical errors here and there, that do however impact the understanding of the document.
My biggest issue with this paper is that in its majority it talks about sorting of rare objects from a blood sample. There are a lot of pages on dielectrophoretic, magnetic, acoustic methods, as well as inertial focusing, deterministic lateral displacement and pinched flow fractionation, all of which are of course well established methods for sorting and isolating rare objects but not really related to the rest of the manuscript about flow cytometry (in the conventional meaning of the word).
The first many pages of the article talk about the lasers used, alignment, cameras etc. This is followed by a very thorough description of cell labeling. The first part is mostly focused on fluorescent particles/dyes etc while in the end some mention is given to magnetic particles or acoustic particles. Until section 2.2 everything that is written sort of suggests a standard flow cytometer. After 2.2 the focus seems to change completely to particle isolation and sorting techniques that have as such nothing to do with the previous part.
I would therefore suggest to the authors to create a much clearer red thread through the manuscript, e.g. by using some of the introduction to clarify what they are going to talk about and why.
Answer: In this review, we aimed to summarize the data on the investigation of rare blood objects with the methods of flow cytometry. In our consideration, the investigation of cells of interest with flow cytometry includes three main steps, which are cell visualization, cell sorting, and cell data analysis. Therefore, we tried to focus on the advances in the methods and approaches of preliminary cell preparation (labeling and isolation) along with advances in flow cytometry analysis including hardware and software development. We have modified the Introduction of the manuscript to make it clearer for the audience as suggested by the Reviewer.
Moreover, there are only 3 figures in the entire article, most of which provide very little information on the subject. Particularly figure 3 in no way conveys any sort of information on dielectrophoretic, magnetic and acoustic methods, other than the fact that all three focus particles one way or the other. Figure 2 is in principle also not very useful. The authors review more or less 350 papers, several of which can be pointed out for some clever solution. I would very much like to see a figure from some of these articles, either by asking for permission to use the figures from the articles or by redrawing the concepts in a new figure. Otherwise the article becomes very cumbersome to read, with too much text.
Answer: We have added additional figures to our manuscript for better illustration of methods and devices we are talking about.
In page 25, line 1165 the authors say that inertia is observed from Reynolds number 0 to 2000. I would probably argue that you cannot have a Reynolds number that is zero.
Answer: We have modified “0 Reynolds number” to “very low Reynolds number”.
Page 31, line 1484: Is it really 3 mm and 6 mm particles?
Answer: The size of the particles in the original article by Nitta et al. was 3 µm and 6 µm. We have corrected this misspelling in the manuscript text.
I suggest the authors change their title to not include "flow cytometry", as I believe it gives the wrong impression. Also, I would try to write something about the requirements necessary for a system like this to be of diagnostic value, in terms of e.g. throughput, analysis time, sensitivity. For some of the described systems, this information is highlighter, but it would be nice to add a table comparing the different techniques. Should the authors insist on talking about flow cytometry, I would also urge them to include a table with some commercial systems and what they are using of different techniques, what they are used for etc.
Answer: We have added a short section to our manuscript describing the advantages and limitations of modern flow cytometry along with the requirements of the studied objects.
I believe that with some of theses changes this paper will be a valuable addition to the literature.
Reviewer 2 Report
The article, "Detection of rare objects by flow cytometry: imaging, cell sorting, and deep learning approaches", is a comprehensive, well-researched and well-written article. It extensively covers biophysical aspects of the flow cytometer to sorting and data analyses.
To further educate readers on this technique, I recommend adding a section on "strengths and weakness of flow cytometry", and downstream applications of a sorted cell-population.
Author Response
To further educate readers on this technique, I recommend adding a section on "strengths and weakness of flow cytometry", and downstream applications of a sorted cell-population.
Answer: we have added a corresponding section to the manuscript as suggested by the Reviewer.
Round 2
Reviewer 1 Report
The authors have addressed my comments. I am particularly happy about the addition of relevant images in the manuscript, which adds relevant information to the article and assists the understanding.
I only have one comment for the authors to address and it concerns the introduction. Although the added paragraph in the beginning motivates the content better than in the original article, there is significant overlapping between the added content (lines 30-38) and the previous content (38-45).
I would suggest keeping the new content (and moving the references 1-5 to this new paragraph), and then skipping lines 38-45 and going directly to lines 45-49, with a bit of rephrasing.
Some of the new text requires some English language editing, but otherwise I find this paper to be suitable for publication.
Author Response
Comment:
I only have one comment for the authors to address and it concerns the introduction. Although the added paragraph in the beginning motivates the content better than in the original article, there is significant overlapping between the added content (lines 30-38) and the previous content (38-45).
I would suggest keeping the new content (and moving the references 1-5 to this new paragraph), and then skipping lines 38-45 and going directly to lines 45-49, with a bit of rephrasing.
Some of the new text requires some English language editing, but otherwise I find this paper to be suitable for publication.
Answer:
We have modified the introduction as suggested by the Reviewer. The modified paragraph is marked with yellow in the manuscript text. Also, we have additionally revised English and corrected some misspellings.